

# SymTFT for (3+1)d gapless SPTs and obstructions to confinement

**Andrea Antinucci[1,2], Christian Copetti[3] and Sakura Schäfer-Nameki[3]**

**1** SISSA, Via Bonomea 265, 34136 Trieste, Italy
**2** INFN, Sezione di Trieste, Via Valerio 2, 34127 Trieste, Italy
**3** Mathematical Institute, University of Oxford, Andrew Wiles Building,
Woodstock Road, Oxford, OX2 6GG, UK

## Abstract

We study gapless phases in (3+1)d in the presence of 1-form and non-invertible duality symmetries. Using the Symmetry Topological Field Theory (SymTFT) approach, we classify the gapless symmetry-protected (gSPT) phases in these setups, with particular focus on intrinsically gSPTs (igSPTs). These are symmetry protected critical points which cannot be deformed to a trivially gapped phase without spontaneously breaking the symmetry. Although these are by now well-known in (1+1)d, we demonstrate their existence in (3+1)d gauge theories. Here, they have a clear physical interpretation in terms of an obstruction to confinement, even though the full 1-form symmetry does not suffer from 't Hooft anomalies. These igSPT phases provide a new way to realize 1-form symmetries in CFTs, that has no analog for gapped phases. The SymTFT approach allows for a direct generalization from invertible symmetries to non-invertible duality symmetries, for which we study gSPT and igSPT phases as well. We accompany these theoretical results with concrete physical examples realizing such phases and explain how obstruction to confinement is detected at the level of symmetric deformations.

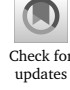
doi:10.21468/SciPostPhys.18.3.114

# 1 Introduction

Once symmetry considerations are taken into account, phases of matter in any spacetime dimensions have a rich and intricate structure, which in its full generality is only starting to get explored. The main new ingredient in recent studies is the inclusion of so-called non-invertible or categorical symmetries in higher dimensions as initiated in [1–3] (for reviews, see [4,5]). In (1+1)d, gapped phases with certain categorical symmetries (notably Tambara-Yamagami categories) have been first discussed in [6,7], while a general proposal combining the standard Landau paradigm of phases with categorical symmetries resulted in the *Categorical Landau paradigm* of [8]. However, for higher-categories a different approach is required. In the next subsection, we will review how gapped and gapless phases with categorical symmetries can be characterized through the lens of the Symmetry Topological Field Theory (SymTFT) [9–13].

While symmetries are a powerful tool in deepening our knowledge of gapped phases, for gapless phases, the constraints are much less stringent, at least if the symmetry is finite. The constraints imposed by categorical symmetries on gapped phases have been explored in depth in (1+1)d [6–8,14–19] and (2+1)d e.g. in [20–26]. An interesting common ground between these two worlds is provided by *(intrinsically) gapless SPTs*, or (i)gSPTs for

short [27–38]. While in this instance the full symmetry $\mathcal{S}$ is anomaly-free[1] it does not act faithfully on the gapless degrees of freedom in the IR. We will denote the faithfully acting quotient by $\mathcal{S}' = \mathcal{S}/\mathcal{S}_{\text{gapped}}$. Gapless SPTs are born from the realization that the symmetry $\mathcal{S}_{\text{gapped}}$, while not acting in the IR, can still be realized in a nontrivial manner.

(Non-intrinsically) gSPTs are relative phases in which two IR theories differ by their decoration by an SPT phase for $\mathcal{S}_{\text{gapped}}$. Although a single gSPT phase can be gapped in a $\mathcal{S}$-symmetric fashion –leading to an SPT phase for the full $\mathcal{S}$– two different gSPTs cannot be connected without a phase transition breaking the $\mathcal{S}$ symmetry. gSPTs have been shown to emerge in many lattice systems and have been described using QFT methods [27–38]. Most of these examples are in (1+1)d. For a discussion of a gSPT phase (for a 0-form symmetry) in (3 + 1)d see [39].

A stronger concept is that of an igSPT. This is a gapless phase whose topological features have no analogue in gapped phases; hence, it cannot be deformed to a gapped SPT in a symmetric way. igSPTs typically arise when $\mathcal{S}$ extends $\mathcal{S}'$ in a nontrivial manner:[2]

$$1 \to \mathcal{S}_{\text{gapped}} \to \mathcal{S} \to \mathcal{S}' \to 1 \,. \tag{1}$$

As shown in [29] this allows the symmetry $\mathcal{S}'$ to be realized in an anomalous fashion. Thus, for a low-energy observer, the gapless degrees of freedom cannot be gapped while preserving the $\mathcal{S}'$ symmetry. In other words, symmetric deformations of IR fixed points can only lead to either gapless phases or SSB phases of $\mathcal{S}'$.[3] Known examples of igSPTs exist in (1+1)d for groups [29, 33], and for non-invertible symmetries [38]. The simplest example is that of $\mathcal{S} = \mathbb{Z}_4$ in (1+1)d. Splitting a $\mathbb{Z}_4$ background field as $A = A' + 2A_{\text{gapped}}$ with $\delta A_{\text{gapped}} = \beta(A')$[4] the igSPT partition function reads:

$$Z_{\text{igSPT}}[A', A_{\text{gapped}}] = \exp\left( \pi i \int A' \cup A_{\text{gapped}} \right) Z_{\text{gapless}}[A'] \,. \tag{2}$$

Clearly, due to the extension, the partition function is gauge invariant under $A' \to A' + d\lambda'$ iff $Z_{\text{gapless}}$ realizes an anomalous $\mathbb{Z}_2$ symmetry.

While igSPTs have been studied mainly in (1+1)d for zero-form symmetries, it is clear that such symmetry extensions should play a similarly important role for higher symmetries in higher dimensions. The purpose of the present paper is to determine which (i)gSPT phases can arise in (3+1)d – whilst also revisiting the (1+1)d case more systematically – from a SymTFT perspective and provide some concrete QFT applications.

Our main result is that igSPT phases also naturally arise in the presence of 1-form symmetries, and they are of clear physical relevance, as they forbid the theory from becoming completely confined by IR relevant deformations. In a similar manner, we will discuss igSPTs for non-invertible duality symmetries, extending the (1 + 1)d results in [38]. Importantly, we will show how the duality symmetry can sometimes make gSPT phases into igSPT ones.

Regarding the (i)gSPT phases for 1-form symmetries, an important remark is in order. Strictly speaking the situation is different from the 0-form symmetry case where the whole symmetry $\mathcal{S}$ is not spontaneously broken: the IR symmetry $\mathcal{S}' = \mathcal{S}/\mathcal{S}_{\text{gapped}}$ acting non-trivially in the CFT is necessarily spontaneously broken, since conformal symmetry is incompatible with area law for line operators. However, there are two distinct ways of breaking a 1-form symmetry spontaneously, either by a topological order in which the line operators are topological in the IR, or –and this is the case under consideration– by a CFT that has non-topological

---

[1]By "anomaly-free" in the present work we mean that the symmetry category $\mathcal{S}$ admits a fiber functor [7].

[2]For categorical symmetries the definition of an extension is somewhat tricky, however the SymTFT will allow us to bypass such subtleties.

[3]The situation is much different if we also allow relevant deformations in the UV to occur. Since the full symmetry $\mathcal{S}$ is anomaly-free these are allowed to gap-out the theory.

[4]$\beta$ is the Bockstein map associated to the sequence $1 \to \mathbb{Z}_2 \to \mathbb{Z}_4 \to \mathbb{Z}_2$ and explicitly $\beta(A') = \delta A'/2$.

conformal lines. The two types of breaking can be intrinsically distinguished, since in the former there is an emergent 2-form symmetry, which is absent in the latter. We will call the second scenario *conformal 1-form symmetry breaking*. When we talk about gSPT phases for a 1-form symmetry, the quotient of the symmetry that acts non-trivially on the gapless sector is spontaneously broken, but in the conformal way. The igSPT phases for 1-form symmetries are obstructions for a CFT with conformal 1-form symmetry breaking to be deformed into a gapped symmetry preserving phase. We now give a brief overview of the SymTFT and of how it can be used to address the problem of characterizing (i)gSPT phases [38].

## 1.1 Gapped and gapless phases with categorical symmetries

The Symmetry Topological Field Theory (SymTFT) [9, 10, 12, 13][5] of a $d$-dimensional theory $\mathcal{T}$ with symmetry $\mathcal{S}$ is obtained by gauging the symmetry in $d+1$ dimensions. The SymTFT for $(1+1)$d theories is the Turaev-Viro TQFT for $\mathcal{S}$ [66]. The central object in this analysis will be played by the topological defects of the SymTFT, which form the Drinfeld center $\mathcal{Z}(\mathcal{S})$. The original theory $\mathcal{T}$ is obtained by placing the SymTFT in an interval with two boundary conditions:

- Gapped symmetry boundary $\mathfrak{B}^{\mathrm{sym}}$: this boundary encodes all the symmetry aspects. In particular, the topological defects on this boundary realize the symmetry $\mathcal{S}$. It is specified in terms of which topological defects can end on this boundary. This will be referred to as a Lagrangian algebra[6] $\mathcal{L}$ of the Drinfeld center $\mathcal{Z}(\mathcal{S})$.

- Physical boundary $\mathfrak{B}^{\mathrm{phys}}$: this not necessarily gapped boundary condition describes how topological objects in $\mathcal{Z}(\mathcal{S})$ couple to dynamical defects of the QFT.

Topological defects, i.e. Drinfeld center elements, that can end on both boundaries correspond to the generalized charges under $\mathcal{S}$ [45].

**SymTFT characterization of gapped phases.** To construct any phase for a categorical symmetry $\mathcal{S}$[7] we specify the symmetry boundary to be $\mathfrak{B}^{\mathrm{sym}} = \mathcal{L}_{\mathcal{S}}$, i.e. the Lagrangian algebra associated to the symmetry $\mathcal{S}$: the algebra specifies the topological defects that can end/condense,[8] and the symmetry defects are those that project onto the symmetry boundary (and are mutually non-local with the defects in the algebra). As proposed in [8, 18, 26, 45], to specify a gapped phase, we pick as the physical boundary any Lagrangian algebra $\mathcal{L}_i$ of $\mathcal{Z}(\mathcal{S})$ (including the one $\mathcal{L}_{\mathcal{S}}$):

$$
\begin{array}{ccc}
\mathfrak{B}^{\mathrm{sym}}_{\mathcal{S}} & \mathfrak{B}^{\mathrm{phys}} = \mathcal{L}_i & \mathcal{T}_i \\
\mathfrak{Z}(\mathcal{S}) & = & \bigg|
\end{array}
\tag{3}
$$

where $\mathcal{T}_i$ denotes the gapped $\mathcal{S}$-symmetric phase obtained by this SymTFT construction.

---

[5]See [8, 38, 40–65] for recent applications of the SymTFT construction, including its extension to continuous symmetries.

[6]Strictly speaking this terminology should be used only in (1+1)d. We will not consider so-called non-minimal boundary conditions here see [26], and thus also use the term Lagrangian to characterize the set of topological defects that end on the gapped boundary.

[7]By this we mean any higher fusion category symmetry.

[8]The converse is not true: there can be two different Lagrangian algebras with the same underlying object but different morphism $\mu: \mathcal{L} \times \mathcal{L} \to \mathcal{L}$. We will not pay attention to this in the following, as this instance never arise in the examples considered.

The order parameters are given by the topological defects that can end on both boundaries, i.e. $\mathcal{L}_{\mathcal{S}} \cap \mathcal{L}_i$ (though for higher categorical symmetries we need to be careful about what we mean by this). In particular, SPT phases are characterized by requiring the intersection to be trivial.

**Gapless phases.** To describe symmetric gapless phases, which, e.g., can be phase transitions between two gapped phases, we follow the approach in [19] (see also [33, 67] for related discussions): first determine the condensable algebras $\mathcal{A}$ of the Drinfeld center, i.e., not necessarily maximal algebras. These define an interface $\mathcal{I}_{\mathcal{A}}$ between the SymTFT for $\mathcal{S}$ and a reduced SymTFT for a symmetry $\mathcal{S}'$. Inserting a physical boundary for the symmetry $\mathcal{S}'$ results in the following "club sandwich" setup:

$$
\begin{array}{ccc}
\mathfrak{B}_{\mathcal{S}}^{\text{sym}} & \mathcal{I}_{\mathcal{A}} & \mathfrak{B}_{\mathcal{S}'}^{\text{phys}} \quad \mathcal{T} \circlearrowleft \mathcal{S} \\
\boxed{\mathfrak{Z}_{d+1}(\mathcal{S}) \;\; \mathfrak{Z}_{d+1}(\mathcal{S}')} & = & |
\end{array}
\tag{4}
$$

This constructs an $\mathcal{S}$-symmetric gapless phase, in which only the symmetry $\mathcal{S}'$ acts faithfully on the gapless degrees of freedom. The construction can also be understood by starting with an $\mathcal{S}'$-symmetric transition between two gapped phases $\mathcal{T}_1'$ and $\mathcal{T}_2'$. The above setup provides a map (functor) to $\mathcal{S}$-symmetric phases, and the transition becomes a gapless phase for $\mathcal{S}$. For more details see [19].

**Symmetry classification of phases.** In [38] it was proposed to organize phases for a categorical symmetry $\mathcal{S}$ in terms of a Hasse diagram, using the partial order given by the condensable algebras. Furthermore, we can define two gaps: the energy gap $\Delta$, which is the standard gap in the spectrum, and the symmetry gap $\Delta_{\mathcal{S}}$. The latter indicates whether charges are realized in the IR or not. If $\Delta_{\mathcal{S}} > 0$ then not all charges are realized in the IR and is set by the energy of the first excited state that carries the charge that is missing in the IR (i.e. the confined charges). Using this, we can now characterize phases as follows:

1. **Gapped (energy gap $\neq$ 0),** specified by a Lagrangian algebra $\mathcal{L}$:

    (a) SSB: spontaneous symmetry breaking phase.

    (b) SPT: symmetry protected phase: $\mathcal{L} \cap \mathcal{L}_{\mathcal{S}} = \{1\}$.

2. **Gapless (energy gap vanishes),** associated to a non-maximal condensable algebra $\mathcal{A}$:

    (a) gSSB: gapless spontaneous symmetry breaking phase. The symmetry $\mathcal{S}'$ is realized on gapless modes while the symmetry $\mathcal{S}_{\text{gapped}}$ is spontaneously broken.

    (b) gSPT: gapless symmetry protected phase: $\mathcal{A} \cap \mathcal{L}_{\mathcal{S}} = \{1\}$.

    (c) igSPT: gSPT that cannot be deformed to a (gapped) SPT for the full symmetry. $\mathcal{A}$ cannot be extended to a Lagrangian algebra $\mathcal{L}$, which gives rise to an SPT.

    (d) igSSB: gSSB that cannot be deformed to an SSB phase while keeping the same number of vacua/universes.

The igSPTs are in particular interesting because they provide examples of critical theories that cannot be deformed without breaking the symmetry. Several examples of this exist in (1+1)d. We will extend this to $(3 + 1)$d inlcuing invertible (1-form) symmetries and non-invertible (0-form) symmetries.

Table 1: SPT, gSPT and igSTP in (1+1)d for abelian 0-form symmetry.

| 0-form Symmetry | Phase type | Algebra |
|---|---|---|
| $\mathbb{Z}_n$ | SPT | — |
| | gSPT | $\mathcal{A}_{\mathbb{B}_p,\psi} : \mathbb{B}_p = \{qx \mid x = 0, \cdots, p-1\},\ n = pq$ $\psi(q) = qm,\ m = rp/\gcd(p,q)$ |
| | igSPT | $\mathcal{A}_{\mathbb{B}_p,\psi} : r \neq 0 \bmod(\gcd(p,q))$ |

## 1.2 Summary of results

Let us give a short summary of our main results. In **section 2** we discuss (i)gSPT phases for a generic abelian group $\mathbb{A}$ in $(1+1)$d. This is meant as a warm-up for the reader to get acquainted with the formalism. Intrinsic phases are described by a subgroup $\mathbb{B} \subset \mathbb{A}$ and an alternating homomorphism $\psi : \mathbb{B} \to \mathbb{A}^\vee$ (namely $\psi(b)b = 1$ for all $b \in \mathbb{B}$), which cannot be extended to an alternating homomorphism $\widehat{\psi}$ over the whole $\mathbb{A}$. Table 1 contains a summary of relevant examples. We then show how, in a physical system, a KT transformation can be used to reach the igSPT phases and embed this in a UV complete example.

In **section 3** we extend the analysis to phases with a 1-form symmetry $\mathbb{A}^{(1)}$ that is a finite abelian group. The extension problem is analogous, but now $\psi : \mathbb{B} \to \mathbb{A}^\vee$ must have a symmetric (as opposite to alternating) property; see table 2 for a recap of our results. We then discuss the KT transformation and its interpretation in terms of the 't Hooft picture for confinement. We also show that in fact the igSPT phase does not admit IR deformations to a trivially gapped phase by examining the allowed deformations, both with fermion masses and with monopole potentials.[9] The physical mechanism at play is remarkably simple: the dyonic operators we need to condense are inaccessible in the IR, as they become confined at a higher energy scale, thus forbidding the existence of the necessary monopole potential driving the theory to a trivially gapped phase.

In **section 4** we use the methods developed in [50] to extend the analysis to duality-type defects in 1+1 and 3+1 dimensions. We find various types of igSPT, which we dub Type I-II-III. A Type I phase has a UV symmetry that describes an igSPT for the invertible part, which is duality invariant. In the IR the invertible part is anomalous, while the duality defect becomes anomaly-free and invertible. A Type II phase describes a duality invariant gSPT for the invertible symmetry. In the IR a non-invertible — but anomalous — duality symmetry acts instead. Finally, a Type III phase only has an anomalous duality symmetry acting in the IR. We then give a concrete example of how such phases cannot be deformed to duality-invariant confined phases.[10] A brief summary of the relevant examples can be found in table 3.

## 2 Classification for abelian 0-form symmetries in (1+1)d

For general finite groups $G$ the possible gSPT and igSPT phases were classified in [33] using the results of [71] on condensable algebras in the SymTFT. Nevertheless, in view of later

---

[9]Here we abuse terminology. By "monopole potential" we mean a potential for local operators stemming from the $S^1$ reduction of the $(3+1)$d dyonic lines.

[10]Duality-invariant SPT phases and RGs leading to them have been studied in [68–70].

Table 2: SPT, gSPT and igSTP in (3+1)d for 1-form symmetry.

| 1-form Symmetry | Phase type | Algebra |
|---|---|---|
| $\mathbb{Z}_n$ | SPT | $\mathcal{L}_r = \{(x, rx) \mid x = 0, \cdots, n-1\}, = 0, \cdots, n-1$ |
| | gSPT | $\mathcal{A}_{\mathbb{B}_p, \psi} : \mathbb{B}_p = \{qx \mid x = 0, \cdots, p-1\}, \; n = pq$ $\psi(qx) = qmx, \; m = 0, \cdots, p-1$ |
| | igSPT | — |
| $\mathbb{Z}_4 \times \mathbb{Z}_2$ | SPT | $\mathcal{L}_{\mathbb{B}, \psi}$ |
| | gSPT | $\mathcal{A}_{\mathbb{Z}_4, \psi}(4 \text{ choices}), \; \mathcal{A}_{\mathbb{Z}_2, \psi}(6 \text{ choices}), \; \mathcal{A}_{\mathbb{Z}_2^2, \psi}(8 \text{ choices})$ |
| | igSPT | $\mathcal{A}_{\mathbb{Z}_2, \psi}$ 2 choices of $\psi_{s_1, 1}, s_1 = 0, 1$ (67) |
| $\mathbb{Z}_4 \times \mathbb{Z}_4$ | SPT | $\mathcal{L}_{\mathbb{B}, \psi}$ |
| | gSPT | $\mathcal{A}_{\mathbb{Z}_2^2, \psi}$ 16 choices of $\psi_{s_1, s_2, r_1, r_2}$ (74) |
| | igSPT | $\mathcal{A}_{\mathbb{Z}_2^2, \psi}$ 8 choices of $\psi_{s_1, s_2, r_1, r_2}, r_1 \neq s_2$ (74) |

generalizations to TY categories and to higher dimensions, we find it useful to provide a direct classification in the abelian group case. Consider in this section an abelian 0-form symmetry $\mathbb{A}^{(0)}$ in (1+1)d.

## 2.1 SymTFT characterization of gapped phases

The SymTFT is the (2+1)d Dijkgraaf-Witten theory for $\mathbb{A}^{(0)}$. In general, a Lagrangian algebra of $\mathcal{Z}(\mathcal{S})$ of a fusion category $\mathcal{S}$ has to satisfy various consistency conditions (see, e.g. the Appendices of [38, 72]). In the present simplified setting, we need a maximal algebra of lines that are mutually local. In particular, these have spin 1.

The Drinfeld center for the 0-form symmetry $\mathbb{A}$, $\mathcal{Z}(\mathsf{Vec}_{\mathbb{A}})$, is isomorphic to $\mathbb{A} \times \mathbb{A}^\vee$. Anyons are lines labeled by $(a, \alpha)$ that are charged under the electric and magnetic symmetries in the bulk. The braiding $\mathcal{B}$ and topological spins $\theta$ of these anyons are

$$\mathcal{B}[(a, \alpha), (b, \beta)] = \beta(a)\alpha(b), \qquad \theta_{(a, \alpha)} = \alpha(a). \tag{5}$$

The canonical Lagrangian algebra corresponding to the symmetry boundary where the $\mathsf{Vec}_{\mathbb{A}}$ symmetry is realized is

$$\mathcal{L}_{\text{sym}} = \left\{ (0, \alpha) \mid \alpha \in \mathbb{A}^\vee \right\}. \tag{6}$$

The quotient

$$\mathcal{S} = \mathbb{A} \times \mathbb{A}^\vee / \mathcal{L}_{\text{sym}} \cong \mathbb{A}, \tag{7}$$

represents the symmetry group, while $\mathcal{L}_{\text{sym}} \cong \mathbb{A}^\vee$ describes all possible irreducible representations (characters) of $\mathbb{A}$. Different choices of Lagrangian algebras for $\mathcal{L}_{sym}$ lead to categorical symmetry which are gauge-related (Morita equivalent) to $\mathsf{Vec}_{\mathbb{A}}$.

Table 3: igSPTs for Duality-type symmetries in $(1+1)$d (including the Tambara-Yamagami (TY) symmetries), above, and $(3+1)$d (including the 3TY generalizations to 3-categories), below.

| Type | Symmetry | Algebra |
|------|----------|---------|
| I | $\text{Vec}_{D_8}$ | $\mathcal{A}_{\mathbb{Z}_2,1,\psi} = \{(2x, 2x) \,|\, x = 0, 1\}$ |
| | $\text{TY}(\mathbb{Z}_9 \times \mathbb{Z}_9, \phi_0, +)$ | $\mathcal{A}_{\mathbb{Z}_3 \times \mathbb{Z}_3, 1, \psi} = \{(3x, 3y; 3y, 3x) \,|\, x, y = 0, 1, 2\}$ |
| III | $\text{Rep}(D_8)$ | $\mathcal{L} = \{(x, y; x, y), \quad x, y = 0, 1\}$ |

| Type | Symmetry | Algebra |
|------|----------|---------|
| I | $3\text{TY}(\mathbb{Z}_9^{(1)} \times \mathbb{Z}_9^{(1)}, \phi_D)$ | $\mathcal{A}_{\mathbb{Z}_3 \times \mathbb{Z}_3, 1, \psi} = \left\{ \Big((3x, 3y); (\psi(3x, 3y))\Big) \,\Big|\, x, y = 0, 1, 2 \right\}$ |
| II | $3\text{TY}(\mathbb{Z}_4^{(1)} \times \mathbb{Z}_4^{(1)}, \phi_O)$ | $\mathcal{A}_0 = \{(0, 0; 0, 0) \oplus (2, 0; 0, 2)\}$ |
| III | $3\text{TY}(\mathbb{Z}_2^{(1)}, 1)$ | $\mathcal{L} = \{(0, 0) \oplus (1, 1)\}$ |

Given a subgroup $\mathbb{B} \subset \mathbb{A}$, denote by $N(\mathbb{B}) \subset \mathbb{A}^\vee$ the subgroup of characters annihilating $\mathbb{B}$

$$N(\mathbb{B}) = \left\{ \beta \in \mathbb{A}^\vee \,\Big|\, \beta(b) = 1, \forall b \in \mathbb{B} \right\} \cong (\mathbb{A}/\mathbb{B})^\vee, \tag{8}$$

where the last isomorphism is a canonical identification. Equally, $\mathbb{A}^\vee/N(\mathbb{B}) \cong \mathbb{B}^\vee$. One can show (see e.g. Appendix B of [50]) that the Lagrangian algebras of $\mathcal{Z}(\text{Vec}_{\mathbb{A}})$ are classified by a choice of subgroup $\mathbb{B} \subset \mathbb{A}$ and a cocycle $\omega \in H^2(\mathbb{B}, U(1))$:

$$\mathcal{L}_{\mathbb{B}, \omega} = \left\{ (b, \beta \psi(b)) \,\Big|\, b \in \mathbb{B}, \beta \in N(\mathbb{B}) \right\}. \tag{9}$$

Here $\psi : \mathbb{B} \to \mathbb{B}^\vee$ is a group homomorphism determined by $\omega \in H^2(\mathbb{B}, U(1))$ as follows. Using the well-known isomorphism [73] between the group of alternating bicharacters and $H^2(\mathbb{B}, U(1))$, we construct the alternating bicharacter $\chi : \mathbb{B} \times \mathbb{B} \to U(1)$:

$$\chi(b, b') = \frac{\omega(b, b')}{\omega(b', b)}. \tag{10}$$

Then $\psi$ is given by

$$\psi(b)b' = \chi(b, b'). \tag{11}$$

**Characterization of SPT phases.** A (gapped) SPT phase is realized in the SymTFT setup, by choosing the physical boundary to be gapped, i.e. given by a Lagrangian algebra $\mathcal{L}_{\mathbb{B}, \omega}$ such that there is no genuine charged operator after the interval compactification. This means that no anyon is allowed to end on both boundaries

$$\mathcal{L}_{\text{sym}} \cap \mathcal{L}_{\mathbb{B}, \omega} = 1. \tag{12}$$

From the realization (9) we see that $\mathcal{L}_{\text{sym}} \cap \mathcal{L}_{\mathbb{B},\omega} = N(\mathbb{B})$. Therefore, SPT phases are obtained by choosing $\mathbb{B} = \mathbb{A}$ and we recover the usual group-cohomology classification of SPTs in terms of $\omega \in H^2(\mathbb{A}, U(1))$ [74].

Similar considerations apply even if $\mathcal{L}_{sym}$ is not of the form (6). For any symmetry associated with $\mathcal{L}_{\text{sym}} = \mathcal{L}_{\mathbb{B},\omega}$ we can repeat the above analysis finding that an SPT for that symmetry is again a Lagrangian $\mathcal{L}_{\mathbb{B}',\omega'}$ satisfying

$$\mathcal{L}_{\mathbb{B},\omega} \cap \mathcal{L}_{\mathbb{B}',\omega'} = 1. \tag{13}$$

## 2.2 Classification of gSPT and igSPT phases

While Lagrangian algebras determine gapped phases, non-maximal condensable algebras define an interface with a reduced topological order [19, 33, 34, 67]. Picking a (generically non-gapped) physical boundary of the reduced topological order, one constructs a generic, not necessarily gapped phase, and hence we refer to these as gapless phases. Some of them are incompatible with a gapped realization and hence are called intrinsically gapless phases. Therefore, gapless phases are classified by condensable algebras $\mathcal{A}$. Intuitively, the anyons in common between $\mathcal{A}$ and $\mathcal{L}_{\text{sym}}$ give rise to topological local operators, describing discrete vacua, while the reduced topological order describes a part of the symmetry which only acts non-trivially on the gapless sector.

### 2.2.1 Condensable algebras of $\mathcal{Z}(\mathsf{Vec}_{\mathbb{A}})$

Condensable algebras $\mathcal{A}$ of $\mathcal{Z}(\mathsf{Vec}_{\mathbb{A}})$ can be parametrized by subgroups made of bosonic lines, but they are not necessarily maximal. We can characterize them similarly to the Lagrangian ones. Consider the projection $\pi_{\mathbb{A}} : \mathbb{A} \times \mathbb{A}^\vee \to \mathbb{A}$ on the first factor, and define $\mathbb{B} := \pi_{\mathbb{A}}(\mathcal{A}) \subset \mathbb{A}$ so that there is a short exact sequence

$$1 \to \ker(\pi_{\mathbb{A}}|_{\mathcal{A}}) \to \mathbb{A} \to \mathbb{B} \to 1. \tag{14}$$

For a generic condensable algebra, $\ker(\pi_{\mathbb{A}}|_{\mathcal{A}})$ is a subgroup of $N(\mathbb{B})$

$$\mathbb{D} = \ker(\pi_{\mathbb{A}}|_{\mathcal{A}}) \subset N(\mathbb{B}). \tag{15}$$

It is convenient to represent $\mathbb{A}^\vee$ as a group extension

$$1 \to \mathbb{D} \to \mathbb{A}^\vee \to \mathbb{A}^\vee/\mathbb{D} \to 1, \tag{16}$$

so that any character is written as a pair $\alpha = \delta\xi$, $\delta \in \mathbb{D}, \xi \in \mathbb{A}^\vee/\mathbb{D}$. Notice that $\mathbb{A}^\vee/\mathbb{D}$ is a group extension of $\mathbb{B}^\vee$ by $N(\mathbb{B})/\mathbb{D}$. The algebra $\mathcal{A}$ can then be represented by

$$\mathcal{A} = \left\{ \left( b, \delta\psi(b) \right) \mid b \in \mathbb{B}, \delta \in \mathbb{D} \right\}, \tag{17}$$

where $\psi : \mathbb{B} \to \mathbb{A}^\vee/\mathbb{D}$ is a group homomorphism. The trivial spin condition translates into

$$\psi(b)b = 1, \quad \forall b \in \mathbb{B}. \tag{18}$$

We conclude that condensable algebras are labelled by triples $(\mathbb{B}, \mathbb{D}, \psi)$ where $\mathbb{B} \subset \mathbb{A}$, $\mathbb{D} \subset N(\mathbb{B})$ and $\psi : \mathbb{B} \to \mathbb{A}^\vee/\mathbb{D}$ is a group-homorphism such that $\psi(b)b = 1$.[11] We denote condensable algebras by

$$\mathcal{A}_{\mathbb{B},\mathbb{D},\psi} : \qquad \mathbb{B} \subset \mathbb{A}, \quad \mathbb{D} \subset N(\mathbb{B}) \subset \mathbb{A}^\vee, \quad \psi : \mathbb{B} \to \mathbb{A}^\vee/\mathbb{D}. \tag{20}$$

---

[11]Notice that it makes sense to evaluate an element (here $\psi(b)$) of $\mathbb{A}^\vee/\mathbb{D}$ on elements of $\mathbb{B}$ because of the short exact sequence

$$1 \to N(\mathbb{B})/\mathbb{D} \to \mathbb{A}^\vee/\mathbb{D} \to \mathbb{B} \to 1. \tag{19}$$

### 2.2.2 gSPT phases

Gapless phases are obtained by condensable but non-maximal algebras, which define interfaces to a reduced topological order – see (4). The set of charges realized on the ground states is given by anyons of $\mathcal{A}$ which can also end on the symmetry boundary. A gapless SPT (gSPT) phase is a gapless phase in which the only charge realized on the vacuum is the trivial one. Hence, the condensable algebra for a gSPT has to satisfy

$$\mathcal{A}_{\text{gSPT}} \cap \mathcal{L}_{\text{sym}} = 1\,. \tag{21}$$

By the classification above of condensable algebras, it follows that $\mathcal{A}_{\mathbb{B},\mathbb{D},\psi} \cap \mathcal{L}_{\text{sym}} = \mathbb{D}$, and the condition (21) is nothing but $\mathbb{D} = 1$. Thus whenever $\mathbb{B} \subsetneq \mathbb{A}$ is a proper subgroup $\mathcal{A}_{\mathbb{B},1,\psi}$ describes a gSPT. In summary gSPT phases are classified by algebras $\mathcal{A}_{\mathbb{B},\psi} \equiv \mathcal{A}_{\mathbb{B},1,\psi}$, and parametrized by the following data:

1. A choice of a proper subgroup $\mathbb{B} \subsetneq \mathbb{A}$.

2. A choice of a group homomorphism $\psi : \mathbb{B} \to \mathbb{A}^\vee$ such that $\psi(b)b = 1$.

### 2.2.3 igSPT phases

A gSPT phase is called *intrinsic* (or igSPT for short) if it cannot be deformed to an ordinary (gapped) SPT phase. This means that the condensable algebra $\mathcal{A}_{\mathbb{B},\psi}$ is not a subalgebra of a Lagrangian algebra corresponding to an SPT. This would be a Lagrangian algebra containing $\mathcal{L}_{\mathbb{B},1,\psi}$ of the form

$$\left\{ (a, \widehat{\psi}(a)) \mid a \in \mathbb{A} \right\}, \tag{22}$$

where $\widehat{\psi} : \mathbb{A} \to \mathbb{A}^\vee$ is a homomorphism such that $\widehat{\psi}(a)a = 1, \forall a \in \mathbb{A}$, and corresponds to a class in $H^2(\mathbb{A}, U(1))$. We conclude that a gSPT classified by $(\mathbb{B}, \psi)$ is an igSPT if and only if $\psi : \mathbb{B} \to \mathbb{A}^\vee$ cannot be extended to a homomorphism $\widehat{\psi} : \mathbb{A} \to \mathbb{A}^\vee$ while preserving the property $\widehat{\psi}(a)a = 1$.

## 2.3 Examples

We now provide several explicit examples of igSPTs in $(1+1)$d. Since we will be dealing with cyclic groups, we will use *additive* notation for ease of reading.

### 2.3.1 Minimal (i)gSPT example: $\mathbb{A} = \mathbb{Z}_4$

The case of $\mathbb{Z}_4$ is very well known [75,76]. Let us nevertheless consider it in the context of our general classification. $\mathbb{Z}_4$ has one non-trivial proper subgroup $\mathbb{B} = \mathbb{Z}_2 = \{0,2\} \subset \mathbb{Z}_4$. There are two possible homomorphisms $\psi : \mathbb{Z}_2 \to \mathbb{Z}_4^\vee$, classified by the choice of possible order-two elements of $\mathbb{Z}_4$ to assign $\psi(2)$: the trivial homomorphism and $\psi(2) = 2$. The first case gives a (nonintrinsic) gSPT. The other possibility also defines a gSPT since $\psi(2)2 = 4 \mod(4) = 0$. The last case is also an igSPT. In fact, the extension of $\psi : \mathbb{Z}_2 \to \mathbb{Z}_4^\vee$ to $\widehat{\psi} : \mathbb{Z}_4 \to \mathbb{Z}_4^\vee$ must satisfy $\widehat{\psi}(2) = 2\widehat{\psi}(1)$, so $\widehat{\psi}(1)$ is 1 or 3, and in both cases $\widehat{\psi}(1)1 \neq 0 \mod(4)$. The igSPT algebra is

$$\mathcal{A}_{\mathbb{Z}_2,1,\psi} = \{(2x, 2x) \mid x = 0, 1\}\,. \tag{23}$$

### 2.3.2 Cyclic groups $\mathbb{A} = \mathbb{Z}_n$

Subgroups $\mathbb{B} \subset \mathbb{Z}_n$ are labelled by divisors $p$ of $n$. Let $n = pq$, then the subgroup is

$$\mathbb{B}_p = \{qx \mid x = 0, \ldots, p-1\} \cong \mathbb{Z}_p\,. \tag{24}$$

Homomorphsims $\psi : \mathbb{B}_p \to \mathbb{Z}_n^\vee$ are determined by picking an order $p$ element $y \in \mathbb{Z}_n^\vee \cong \mathbb{Z}_n$[12] and declaring that $\psi(q) = y$. Clearly the elements of order $p$ of $\mathbb{Z}_n^\vee$ forms the subgroup

$$N(\mathbb{B}_q) = \{qm \mid m = 0, \ldots, p-1\} \cong \left(\mathbb{Z}_n/\mathbb{B}_q\right)^\vee \cong \mathbb{Z}_p \,, \tag{25}$$

hence we have $p$ possible homomorphisms $\psi_m : \mathbb{B}_p \to \mathbb{Z}_n^\vee$ labelled by $m \in \mathbb{Z}_p$, and defined by

$$\psi_m(q) = qm \,. \tag{26}$$

We see that

$$\psi_m(qx)\,qx = q^2 x^2 m \bmod(n)\,, \tag{27}$$

so that this is alternating if and only if $p$ divides $qm$. This implies that $m$ must be proportional to $p/\gcd(p,q)$, namely it takes value in the subgroup $\mathbb{Z}_{\gcd(p,q)} \subset \mathbb{Z}_p$. This subgroup classifies the gSPT phases for $\mathbb{Z}_n$.

Let us consider the igSPT phases: there is only a trivial SPT phase for these symmetries, and these algebras for the igSPT are not contained within this. Let us show this in a way which is helpful in other cases.[13] We should ask when $\psi_m$ with $m = pr/\gcd(p,q)$ can be extended to $\widehat{\psi}_m : \mathbb{Z}_n \to \mathbb{Z}_n^\vee$ in such a way that $\widehat{\psi}_m(a)a = 0 \bmod(n)$ for all $a \in \mathbb{Z}_n$. Notice that $q\widehat{\psi}_m(1) = qm \bmod(n)$, so that

$$\widehat{\psi}_m(1) = m + kp\,, \ k = 0, \ldots, q-1\,. \tag{28}$$

Thus the condition $\widehat{\psi}_m(1)1 = 0 \bmod(n)$ becomes

$$m + kp = 0 \bmod(n)\,. \tag{29}$$

Clearly this can never be satisfied by a non-trivial $m$. Thus we conclude that any nontrivial $\mathbb{Z}_n$ gSPT phase is also an igSPT. The igSPT algebras are

$$\mathcal{A}_{\mathbb{Z}_p,1,\psi_m} = \{(qx, mqx) \mid x = 0, \ldots, p-1\}\,, \ m = pk/\gcd(p,q)\,, \ k \in \mathbb{Z}_{\gcd(p,q)}\,, \ k \neq 0\,. \tag{30}$$

To summarize, the gSPT phases for $\mathbb{Z}_n$ are classified by divisors $p$ of $n$ and an element in $\mathbb{Z}_{\gcd(p,q)}$. Hence we have

$$G(n) = \sum_{p\mid n} \gcd(p,q)\,, \tag{31}$$

many gSPT phases.

### 2.3.3 igSPTs for $\mathbb{A} = \mathbb{Z}_n \times \mathbb{Z}_n$

We do not attempt to classify all gapless phases for $\mathbb{A} = \mathbb{Z}_n \times \mathbb{Z}_n$, rather we focus on igSPTs existing for $n = pq$, $\gcd(p,q) \neq 1$, which are representative of the general scenario. This is instructive since $\mathbb{Z}_n \times \mathbb{Z}_n$ also admits gapped SPTs, so not all non-trivial gSPTs are automatically intrinsic.

We look at the subgroup $\mathbb{B} = \{(qx, qy)\} \cong \mathbb{Z}_p \times \mathbb{Z}_p$. The most general homomorphism $\mathbb{Z}_p \times \mathbb{Z}_p \to \mathbb{Z}_n \times \mathbb{Z}_n$ is determined by four numbers $\bmod(p)$

$$\psi_{s_1,s_2,r_1,r_2}(qx, qy) = (s_1 qx + r_1 qy, s_2 qx + r_2 qy)\,, \tag{32}$$

and this is alternating if and only if $s_1, r_2, r_1 + s_2$ are proportional to $p/\gcd(p,q)$. However, it admits an alternating extension $\widehat{\psi} : \mathbb{Z}_n \times \mathbb{Z}_n \to \mathbb{Z}_n \times \mathbb{Z}_n$ only if $s_1 = r_2 = r_1 + s_2 = 0 \bmod(p)$,

---

[12]We pick the (non-canonical) isomorphism $\mathbb{Z}_n \to \mathbb{Z}_n^\vee$ that assign to $a \in \mathbb{Z}_n$ the character $\chi_a(b) = \exp\left(\frac{2\pi iab}{n}\right)$.

[13]Either for $\mathbb{A} = \mathbb{Z}_n \times \mathbb{Z}_n$ or, as will be mostly important for us, in (3+1)d.

the value of $r \equiv r_1 = -s_2 = 0, \ldots, n-1$ (this is a lift of $r_1 = -s_2$ to $\mathbb{Z}_n$) being the value of the ordinary gapped SPT classified by $H^2(\mathbb{Z}_n \times \mathbb{Z}_n, U(1)) = \mathbb{Z}_n$.

Hence for $\gcd(p,q) \neq 1$ we can choose $s_1, r_2, r_1 + s_2$ to be non-trivial, producing igSPT phases. We have $p\left(\gcd(p,q)^3 - 1\right)$ such phases. Those with $s_1, r_2$ non-trivial but $r_1 + s_2 = 0$ are the igSPTs for the two $\mathbb{Z}_n$ factors that we already discussed, while $r_1 + s_2 = kp/\gcd(p,q)$, $k \neq 0$ are new. For example, for $n = 9$, an igSPT is given by $p = q = 3$, $s_1 = r_2 = 0$, $r_1 = s_2 = 1$. The algebra is[14]

$$\mathcal{A}_{\mathbb{Z}_3 \times \mathbb{Z}_3, 1, \psi_{0,1,1,0}} = \{(3x, 3y; 3y, 3x) \,|\, a, b = 0, 1, 2\}\,. \tag{33}$$

More generally, letting $\ell = p/\gcd(p,q)$, we denote these algebras as $\mathcal{A}_{\mathbb{Z}_p \times \mathbb{Z}_p, 1, \psi}$ with

$$\psi(qx, qy) = \left(k_1 \ell qx + (r + k_2 \ell)qy\,, -rqx + k_3 \ell qy\right), \tag{34}$$

determined by $k_1, k_2, k_3 \in \mathbb{Z}_{\gcd(p,q)}$ and $r \in \mathbb{Z}_p$. These define a class of igSPTs for $\mathbb{Z}_n \times \mathbb{Z}_n$.

## 2.4 Physical construction of igSPTs

In this section we describe how to use a KT transformation to generate examples of igSPTs, following [35, 77]. We then describe the effect of embedding their construction into a UV complete QFT.

### 2.4.1 KT transformation to igSPT phases

We will now illustrate these (i)gSPTs with concrete (1+1)d Field theories. This is a slight generalization of the results in [77], which treated the case $\mathbb{A} = \mathbb{Z}_4$. The procedure works in four steps:

1. Consider a theory $\mathfrak{T}_0$ with zero-form symmetry $\mathbb{Z}_n^{(0)}$ (here $n = pq$), gauge $\mathbb{Z}_p \subset \mathbb{Z}_n^{(0)}$ to construct $\mathfrak{T}_0/\mathbb{Z}_p$.

2. Stack a trivial theory with $\mathbb{Z}_q'$ symmetry.

3. Gauge $\mathbb{Z}_p^\vee \times \mathbb{Z}_q'$, with a discrete torsion class $m \in H^2(\mathbb{Z}_p \times \mathbb{Z}_q, U(1)) = \mathbb{Z}_{\gcd(p,q)}$. This produces a theory with $\mathbb{Z}_n \times \mathbb{Z}_q$ symmetry.

4. Identify the gauge field for $\mathbb{Z}_q$ with that for the $\mathbb{Z}_q$ quotient in $\mathbb{Z}_n$.

This will prove instrumental in discussing examples with 1-form symmetry in section 3. Let us now give some further details.

As we have seen, an igSPT for $\mathbb{Z}_n$ is given by a subgroup $\mathbb{Z}_p \subset \mathbb{Z}_n$ such that

$$\gcd(p,q) \neq 1\,, \quad n = pq\,, \tag{35}$$

together with the choice of $m \in \mathbb{Z}_{\gcd(p,q)}$ (the igSPT is non-trivial if $m \neq 1$). This second choice can be understood as an element

$$m \in H^2(\mathbb{Z}_p \times \mathbb{Z}_q, U(1)) = \mathbb{Z}_{\gcd(p,q)}\,, \tag{36}$$

namely an SPT phase for $\mathbb{Z}_p \times \mathbb{Z}_q$. The authors of [76] gave a construction of a continuum QFT that realizes the igSPT for $\mathbb{Z}_4$. We can generalize this construction to produce a continuum QFT realizing of all the igSPT phases that we have classified.

---

[14]Our notation is $(e_1, e_2; m_1, m_2)$, where $e_{1,2}, m_{1,2}$ label the electric and magnetic lines of the two $\mathbb{Z}_n$ groups, respectively.

Table 4: Groups and background fields.

| Group | $\mathbb{Z}_p$ | $\mathbb{Z}_p^\vee$ | $\mathbb{Z}_p^{\vee\vee}$ | $\mathbb{Z}_q$ | $\mathbb{Z}_q'$ | $\mathbb{Z}'^\vee_q$ |
|---|---|---|---|---|---|---|
| Background Field | $A_p$ | $\widehat{A}_p$ | $\widetilde{A}_p$ | $B_q$ | $B_q'$ | $\widehat{B'}_q$ |

We start from any 2d CFT $\mathfrak{T}_0$ with $\mathbb{Z}_n^{\text{in}}$ symmetry. We gauge the subgroup $\mathbb{Z}_p \subset \mathbb{Z}_n^{\text{in}}$, producing a theory $\mathfrak{T}_0/\mathbb{Z}_p$. Since $\gcd(p,q) \neq 1$ the sequence

$$1 \to \mathbb{Z}_p \to \mathbb{Z}_n^{\text{in}} \to \mathbb{Z}_q \to 1, \tag{37}$$

does not split, hence the resulting theory has symmetry $\mathbb{Z}_p^\vee \times \mathbb{Z}_q$ with mixed anomaly [78]

$$\frac{2\pi i}{p} \int_{X_3} \widehat{A}_p \cup \beta(B_q), \tag{38}$$

where $\widehat{A}_p$ is the background field for $\mathbb{Z}_p^\vee \cong \mathbb{Z}_p$, $B_q$ a background for $\mathbb{Z}_q = \mathbb{Z}_n^{\text{in}}/\mathbb{Z}_p$, and $\beta : H^1(X, \mathbb{Z}_q) \to H^2(X, \mathbb{Z}_p)$ is the Bockstein associated with the sequence (37).[15]

Clearly, if we now make $\widehat{A}_p$ dynamical, we recover the theory $\mathfrak{T}_0$. Instead, we perform a slightly different operation. First we declare that the system has a further trivially acting $\mathbb{Z}_q$ symmetry that we denote by $\mathbb{Z}_q'$ to distinguish it from the quotient $\mathbb{Z}_q = \mathbb{Z}_n^{\text{in}}/\mathbb{Z}_p$. This can be thought of as stacking a decoupled trivially gapped system with a $\mathbb{Z}_q'$ symmetry. Denote the background field for this by $B_q'$. The notation for various groups and background fields is summarized in table 4.

We then gauge $\mathbb{Z}_p^\vee \times \mathbb{Z}_q'$, but crucially adding a discrete torsion

$$\exp\left( \frac{2\pi i m}{\gcd(p,q)} \int_{X_2} \widehat{A}_p \cup B_q' \right), \qquad m \in H^2(\mathbb{Z}_p \times \mathbb{Z}_q', U(1)) = \mathbb{Z}_{\gcd(p,q)}. \tag{39}$$

Denoting by $\widetilde{A}_p$ and $\widehat{B'}_q$ the backgrounds for the two dual symmetries $(\mathbb{Z}_p^\vee)^\vee \cong \mathbb{Z}_p$, $(\mathbb{Z}_q')^\vee$ that arise from this gauging, the resulting partition function is

$$Z_{\mathfrak{T}}[\widetilde{A}_p, \widehat{B'}_q, B_q] = \sum_{\widehat{A}_p, B_q'} \exp\left( \frac{2\pi i m}{\gcd(p,q)} \int_{X_2} \widehat{A}_p \cup B_q' + \frac{2\pi i}{p} \int_{X_2} \widetilde{A}_p \cup \widehat{A}_p + \frac{2\pi i}{q} \int_{X_2} \widehat{B'}_q \cup B_q' \right)$$
$$\times Z_{\mathfrak{T}_0/\mathbb{Z}_p}[\widehat{A}_p, B_q]. \tag{40}$$

To obtain the correct cocycle conditions of the new backgrounds, we need to impose gauge invariance under both $\widehat{A}_p \to \widehat{A}_p + \delta\lambda$ and $B_q' \to B_q' + \delta\eta$. This imposes

$$\delta\widehat{B'}_q = 0, \quad \delta\widetilde{A}_p = \beta(B_q). \tag{41}$$

The dual symmetry $\mathbb{Z}_p^{\vee\vee} = \mathbb{Z}_p$ now extends $\mathbb{Z}_q$ producing again a new $\mathbb{Z}_n$ symmetry. This is to be distinguished from the original $\mathbb{Z}_n^{\text{in}}$, as it is not faithfully acting: as we will see shortly

---

[15]In our case it is given explicitly by $\beta(B_q) = \frac{\delta B_q}{p}$ where $B_q$ in the right hand side is an arbitrary lift to $\mathbb{Z}_n$ of the $\mathbb{Z}_q$ gauge field.

the $\mathbb{Z}_p$ subgroup does not act at all. To show this fact let us manipulate the partition function. Rewriting $Z_{\mathfrak{T}_0/\mathbb{Z}_p}$ in terms of $Z_{\mathfrak{T}_0}$, we can perform the sum over $\widehat{A}_p$, that imposes a delta function, that can be solved by the sum over $A_p$, and we remain with

$$Z_{\mathfrak{T}}[\widetilde{A}_p, \widehat{B}'_q, B_q] = \sum_{B'_q} \exp\left(\frac{2\pi i}{q}\int_{X_2} \widehat{B}'_q \cup B'_q\right) Z_{\mathfrak{T}_0}\left[q\left(\widetilde{A}_p + m\frac{p}{\gcd(p,q)}B'_q\right) + B_q\right]. \tag{42}$$

To gain some intuition about the result, consider the particular case $q = p$, $m = 1$ (for $p = 2$ this is the case discussed in [79]). We omit the subscripts. In the sum over $B'$ we can shift $B' \mapsto B' - \widetilde{A}$ to eliminate it from $Z_{\mathfrak{T}_0}$. Hence, this shifted sum over $B'$ reproduces the partition function of $Z_{\mathfrak{T}_0/\mathbb{Z}_p}[\widehat{B'}, A]$, but there is an addition phase factor:

$$Z_{\mathfrak{T}}[\widetilde{A}, \widehat{B'}, B] = \exp\left(-\frac{2\pi i}{p}\int_{X_2} \widehat{B'} \cup \widetilde{A}\right) Z_{\mathfrak{T}_0/\mathbb{Z}_p}[\widehat{B'}, B]. \tag{43}$$

Naively, this seems like stacking a CFT $\mathfrak{T}_0/\mathbb{Z}_p$ and an invertible phase, but it's more nuanced. The theory has a symmetry $\mathbb{Z}_{p^2} \times \mathbb{Z}_p$ with backgrounds $p\widetilde{A}+B$ and $\widehat{B}$, respectively. Focusing on the gapless sector $\mathfrak{T}_0/\mathbb{Z}_p$, the $\mathbb{Z}_p \subset \mathbb{Z}_{p^2}$ does not act, leaving a faithful $\mathbb{Z}_p \times \mathbb{Z}_p$ symmetry (with the first factor being $\mathbb{Z}_{p^2}/\mathbb{Z}_p$) and a mixed anomaly: a gauge transformation $\widehat{B'} \to \widehat{B'} + \delta\lambda$ multiplies $Z_{\mathfrak{T}_0/\mathbb{Z}_p}$ by a phase. Including the symmetry $\mathbb{Z}_p$ with background $\widetilde{A}$ acting trivially, the anomaly is cancelled by the Green-Schwarz mechanism:

$$\exp\left(-\frac{2\pi i}{p}\int_{X_2} \widehat{B'} \cup \widetilde{A}\right) \to \exp\left(-\frac{2\pi i}{p}\int_{X_2} \delta\lambda \cup \widetilde{A}\right) = \exp\left(\frac{2\pi i}{p}\int_{X_2} \lambda \cup \beta(B)\right).$$

Physically, for the CFT $\mathfrak{T}_0/\mathbb{Z}_p$, the anomaly implies that the system cannot be gapped while preserving $\mathbb{Z}_p \times \mathbb{Z}_p$. However, since the anomaly is canceled by a symmetry acting on an invertible phase, by realizing the latter as the IR of a trivially gapped theory, the anomaly is absent in the full theory. From an IR viewpoint, this can be seen as an example of the general story of [80]. Thus, in the full theory, there is no obstruction to gapping it. At low energy, without considering the additional symmetry, we cannot gap the system unless the additional degrees of freedom become massless, closing the symmetry gap, thus encountering a phase transition and allowing further deformations to gap the theory.

### 2.4.2 Embedding into a UV theory

This construction neatly describes the IR phase. To embed it into a full-fledged UV theory, we simply consider the product of theory $\mathfrak{T}_0$ — or an arbitrary QFT flowing to it— with a theory $\mathfrak{T}'$ flowing to a trivially gapped phase and carrying the $\mathbb{Z}'_q$ symmetry action. Let us focus on the case $n = p^2$ and give a concrete example. Consider $\mathfrak{T}_0$ to be a free compact scalar $X$ of radius $R$ and $\mathfrak{T}'$ to be a compact scalar $X'$ of large radius $R'$ deformed by $\lambda\cos(X')$.[16] We take the radius $R'$ large enough so that this term is relevant and the embedding of symmetries is as in table 5.

Stacking the $\mathbb{Z}_p^\vee \times \mathbb{Z}_p'$ SPT $\exp\left(\frac{2\pi i}{p}\int \widehat{A}_p \cup A'_p\right)$ gives charge to twist fields according to table 5.[17] The charges can be used to describe the local field content of the theory after performing

---

[16]We normalize the fields so that the periodicity is always $2\pi$ to simplify the notation.

[17]A simple derivation of this fact, following e.g. [44], is to realize a twist defect through an open background $\delta A = u$, with $u$ the charge of the twisted sector. Performing a gauge transformation and using the SPT action with this background gives non-trivial charges of twist defects.

Table 5: Embedding of symmetries, relevant twist fields and their charge after stacking the SPT for the free boson example.

| Group | $\mathbb{Z}_n$ | $\widehat{\mathbb{Z}}_p$ | $\mathbb{Z}_p$ | $\mathbb{Z}'_p$ |
|---|---|---|---|---|
| Embedding | $\mathbb{Z}_n^m$ | $\widehat{\mathbb{Z}}_p^w$ | $\mathbb{Z}_n^m/\mathbb{Z}_p^m$ | $\mathbb{Z}_p^{w'}$ |
| Twist field | $W = e^{i\widetilde{X}/p^2}$ | $\widehat{V} = e^{iX}$ | $W = e^{i\widetilde{X}/p^2}$ | $V' = e^{iX'/p}$ |

| | $\widehat{\mathbb{Z}}_p$ | $\mathbb{Z}_p$ | $\mathbb{Z}'_p$ |
|---|---|---|---|
| $\widehat{V}$ | 1 | $e^{-2\pi i/p^2}$ | $e^{-2\pi i/p}$ |
| $V'$ | $e^{2\pi i/p}$ | 1 | 1 |
| $W$ | $e^{2\pi i/p^2}$ | 1 | 1 |

the gauging procedure. Surprisingly, we learn that neither $X$ nor $X'$ are well defined field anymore, but rather we should consider:

$$Y = X + X'/p, \quad Z = X' - \widetilde{X}. \tag{44}$$

Written in terms of these fields the cosine potential becomes:

$$\lambda \cos\left(Z + \widetilde{Y}\right), \tag{45}$$

which pins the momentum modes of one field to the winding modes of the other. The faithfully acting (unbroken) symmetry on the IR scalar is just the $\mathbb{Z}_p$ diagonal between momentum and winding, which is anomalous.

# 3 (i)gSPTs for 1-form symmetries in (3+1)d

The formalism we developed in (1+1)d can be adapted to discuss phases – gapped and gapless – with 1-form symmetries in (3+1)d. We carry out a SymTFT analysis, showing the existence – and providing a classification – of igSPT phases for 1-form symmetries. This refines the standard classification of phases of gauge theories. The SymTFT approach also aids in the construction of concrete physical examples, and we provide an interpretation of these phases as topological obstructions to confinement.

## 3.1 SymTFT and gapped phases for 1-form symmetries

The SymTFT for a 1-form symmetry $\mathbb{A}^{(1)}$ in (3+1)d is a five-dimensional TQFT whose topological defects are surfaces that form a group $\mathcal{Z}(\mathbb{A}^{(1)}) = \mathbb{A} \times \mathbb{A}^{\vee}$ governing their fusion. The braiding is anti-symmetric

$$\mathcal{B}((a,\alpha),(b,\beta)) = \alpha(b)\beta(a)^{-1}. \tag{46}$$

The canonical Lagrangian algebra that leads to the 1-form symmetry $\mathbb{A}$ is

$$\mathcal{L}_{\text{sym}} = \left\{(0,\alpha) \mid \alpha \in \mathbb{A}^{\vee}\right\}. \tag{47}$$

Everything is very similar to the (1+1)d case, with the only (but crucial) difference that the braiding is anti-symmetric.

For simplicity, we assume all manifolds are spin. This technical assumption serves to identify different global variants with the same 1-form symmetry and the same choice of charges, but where the line operators have different statistics [81]. The group of discrete torsions

$$H^4(B^2\mathbb{A}, U(1)) \cong \{q : \mathbb{A} \to U(1), \text{ quadratic form}\}, \tag{48}$$

is a central extension of the group of symmetric bilinear forms $\text{Symm}(\mathbb{A})$. The projection map is the polarization

$$\chi_q(a, b) = q(a + b) - q(a) - q(b). \tag{49}$$

The fiber is isomorphic to $\text{Hom}(\mathbb{A}, \mathbb{Z}_2)$, and its elements are called characteristic elements. From a quadratic form $q$ and a gauge field $B \in H^2(X_4, \mathbb{A})$ one produces the discrete torsion

$$\int_{X_4} q(B). \tag{50}$$

The key fact is that different quadratic forms with same polarization give the same integral on any *spin* 4-manifold. Hence we can mod-out $\text{Hom}(\mathbb{A}, \mathbb{Z}_2)$ and work with symmetric bicharacters $\chi \in \text{Symm}(\mathbb{A})$.

As in (1+1)d, Lagrangian algebras of $\mathcal{Z}(\mathbb{A}^{(1)})$ are classified by a subgroup $\mathbb{B} \subset \mathbb{A}$ and a symmetric homomorphism $\psi : \mathbb{B} \to \mathbb{B}^\vee$ as

$$\mathcal{L}_{\mathbb{B},\psi} = \{(b, \beta\psi(b)) \mid b \in \mathbb{B}, \beta \in N(\mathbb{B})\}. \tag{51}$$

The symmetric condition means that $\chi : \mathbb{B} \times \mathbb{B} \to U(1)$, $\chi(b, b') = \psi(b)b'$ is a symmetric bicharacter, and this ensures that the elements of $\mathcal{L}_{\mathbb{B},\psi}$ do not braid among themselves. It also identifies $\psi$ with a discrete torsion element in $H^4(B^2\mathbb{A}, U(1))/\text{Hom}(\mathbb{A}, \mathbb{Z}_2) \cong \text{Symm}(\mathbb{A})$.

In this subsection, we discuss gapped phases with 1-form symmetry; hence let us make some general comment on them. In the UV there are line operators labeled by their charges valued in $\mathbb{A}^\vee$. Since the phase is gapped, at low energy for each line there are two possibilities: either it has area low and flows to the trivial line (confined lines), or it has perimeter law and flows to a non-trivial topological line (deconfined lines). According to 't Hooft a phase is described by a Lagrangian lattice of dyons [82]. More explicitly, given the algebra $\mathcal{L}_{\mathbb{B},\psi}$, perimeter law is assigned to the lines:

$$W^\beta, \quad T^b W^{\psi(b)}, \tag{52}$$

where $T, W$ are Wilson and 't Hooft lines for the universal cover of the gauge group and we only indicate their 1-form symmetry charge through our notation. The set of charges of deconfined lines forms a subgroup $\mathbb{D} \subset \mathbb{A}^\vee$, while $\mathbb{A}^\vee/\mathbb{D}$ is the quotient that labels the confined lines. Its Pontryagin dual $\mathbb{B} = (\mathbb{A}^\vee/\mathbb{D})^\vee$ is the preserved subgroup of the 1-form symmetry, while the quotient $\mathbb{A}/\mathbb{B}$ is spontaneously broken.

We can make this discussion more systematic using the SymTFT approach. Gapped phases with 1-form symmetry $\mathbb{A}^{(1)}$ are classified by Lagrangian algebras $\mathcal{L}_{\mathbb{B},\psi}$. The physical interpretation is the following. Fixing the symmetry to be $\mathbb{A}^{(1)}$ means that the symmetry boundary is determined by $\mathcal{L}_{\text{sym}} = \{(0, \alpha) \mid \alpha \in \mathbb{A}^\vee\}$, hence the symmetry operators are the surfaces $(a, 0) \in \mathcal{Z}(\mathbb{A}^{(1)})$ pushed at the boundary. Looking for gapped phases means that the physical boundary is also topological and determined by a Lagrangian algebra. The surfaces that can end on *both* boundaries give rise to non-trivial topological line operators, namely the deconfined lines. They form the group

$$\mathcal{L}_{\text{sym}} \cap \mathcal{L}_{\mathbb{B},\psi} \cong \mathbb{D} = N(\mathbb{B}). \tag{53}$$

Hence deconfined lines are completely transparent under the subgroup $\mathbb{B} \subset \mathbb{A}$ of the 1-form symmetry, while are detected by the quotient $\mathbb{A}/\mathbb{B}$. The group of non-trivial lines $N(\mathbb{B})$ is the Pontryagin dual of $\mathbb{A}/\mathbb{B}$, representing its set of charges. Moreover $\mathbb{A}^\vee/N(\mathbb{B}) \cong \mathbb{B}^\vee$ is the set of confined lines and is the dual of the trivialized subgroup of the 1-form symmetry.

The presence of $\mathbb{B}$ may also be detected by looking at the twisted sectors. These arise because some surfaces $(b, \psi(b)) \in \mathcal{L}_{\mathbb{B},\psi}$ cannot end on $\mathcal{L}_{\text{sym}}$ and produce non-genuine lines in the twisted sector of $b \in \mathbb{B}$. Importantly, these non-genuine lines are also in general charged under the subgroup $\mathbb{B}$ of the 1-form symmetry $\mathbb{A}^{(1)}$: passing a surface labelled by $b' \in \mathbb{B}$ through a non-genuine line $(b, \psi(b))$ we pick a phase

$$\psi(b)b' = \chi(b, b'). \tag{54}$$

We conclude that the (3+1)d gapped phase is the spontaneous breaking of $\mathbb{A}$ down to $\mathbb{B}$ whose SPT phase is $\psi$. In particular, the phases with $\mathbb{B} = \mathbb{A}$ are SPT phases for the whole 1-form symmetry, and are determined by $\psi$. Thus, we recover the usual classification by $H^4(B^2\mathbb{A}, U(1))$ [83] (more precisely by $\text{Symm}(\mathbb{A})$ on spin manifolds).

As an example, consider $\mathbb{A} = \mathbb{Z}_n$. Subgroups are given by divisors $p|n$:

$$\mathbb{B}_p = \{qx \mid x = 0, \ldots, p-1\} \cong \mathbb{Z}_p, \quad n = pq. \tag{55}$$

Identifying $\mathbb{Z}_n^\vee \cong \mathbb{Z}_n$ we have

$$N(\mathbb{B}_p) = \{py \mid y = 0, \ldots, q-1\} \cong \mathbb{Z}_q, \tag{56}$$

while

$$\mathbb{B}_p^\vee \cong \mathbb{A}/N(\mathbb{B}_p) = \{x \sim x + p \mid x = 0, \ldots, p-1\}. \tag{57}$$

A homomorphism $\psi : \mathbb{B}_q \to \mathbb{B}_q^\vee$ is the multiplication by a number $r = 0, \ldots, p-1$, and is automatically symmetric. Therefore

$$\mathcal{L}_{p,r} = \{(qx, rx + py) \mid x = 0, \ldots, p-1, \ y = 0, \ldots, q-1\}. \tag{58}$$

SPT phases are obtained by setting $p = n$ and are classified by $r \in \mathbb{Z}_n$, i.e.

$$\text{SPT}: \quad \mathcal{L}_r = \{(x, rx) \mid x = 0, \ldots, n-1\}, \quad r = 0, \ldots, n-1. \tag{59}$$

## 3.2 (i)gSPT phases protected by 1-form symmetries

Now we look at gapless phases with 1-form symmetry $\mathbb{A}^{(1)}$. Each line operator, labeled by $\alpha \in \mathbb{A}^\vee$ can either flow to a trivial line, or to a non-trivial line, and the latter form a subgroup $N(\mathbb{B}) \subset \mathbb{A}^\vee$ (hence $\mathbb{B} \subset \mathbb{A}$ is trivial in the IR). However, differently from gapped phases, among the non-trivial lines, some can be topological, while others are not. The first set forms a subgroup $\mathbb{D} \subset N(\mathbb{B})$, and the presence of a gapless sector is characterized by the non-triviality of the quotient $N(\mathbb{B})/\mathbb{D}$ that labels the charges of non-topological lines of the gapless sector. Notice that all non-trivial lines must have perimeter law, hence the symmetry $N(\mathbb{B})^\vee$ under which they are charged is spontaneously broken. However, the symmetry $(N(\mathbb{B})/\mathbb{D})^\vee$ under which the non-topological IR lines are charged is conformally broken, and not associated with an emergent 2-form symmetry.

All of this can be formalized considering non-maximal condensable algebras of the SymTFT. The classification is as follows: the condensable algebras[18] are

- A subgroup $\mathbb{B} \subset \mathbb{A}$.

---

[18]We note that this reference to algebra is not quite accurate as these defects form a sub-higher-category, but we refrain from this in this context unnecessary embellishment in terminology.

- A subgroup $\mathbb{D} \subset N(\mathbb{B})$.

- A group homomorphism $\psi : \mathbb{B} \to \mathbb{A}^{\vee}/\mathbb{D}$ with the property that

$$\psi(b)b' = \psi(b')b, \quad \forall b, b' \in \mathbb{B}. \tag{60}$$

The corresponding algebra is

$$\mathcal{A}_{\mathbb{B},\mathbb{D},\psi} = \left\{ (b, \delta\,\psi(b)) \,\big|\, b \in \mathbb{B}, \delta \in \mathbb{D} \right\}. \tag{61}$$

In general, non-maximal condensable algebras describe gapless phases. The interpretation of (61) is clear from the "Club-Sandwich" picture:

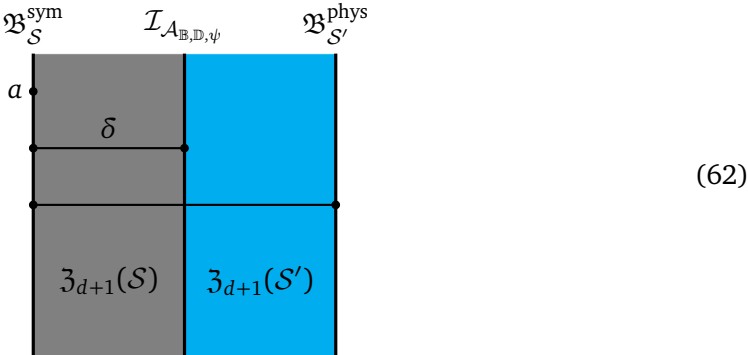

$$\tag{62}$$

Condensing $\mathcal{A}_{\mathbb{B},\mathbb{D},\psi}$ in half space we construct an interface between $\mathfrak{Z}(\mathcal{S})$ and a reduced topological order $\mathfrak{Z}(\mathcal{S}')$, with $\mathcal{S} = \mathbb{A}^{(1)}$. After interval compactification, we can distinguish three types of object, represented in the figure above from bottom to top:

1. Dynamical degrees of freedom charged under $\mathcal{S}'$, described by surface operators extending throughout the bulk.

2. Topological lines $(0, \delta)$ ending both on the $\mathcal{I}_{\mathcal{A}_{\mathbb{B},\mathbb{D},\psi}}$ interface and the symmetry boundary $\mathfrak{B}^{\mathrm{sym}}$. These describe a (deconfined) SSB phase dressing the gapless degrees of freedom.

3. Topological surfaces labelled by $a \in \mathbb{A}$ confined on $\mathfrak{B}^{\mathrm{sym}}$ and describing the full symmetry.

Up to this point everything is the same as in the gapped case. The difference is that now the quotient $\mathbb{A}/\mathbb{B}$ does not act faithfully on topological lines:

$$\mathcal{L}_{\mathrm{sym}} \cap \mathcal{A}_{\mathbb{B},\mathbb{D},\psi} \cong \mathbb{D}, \tag{63}$$

which is in general smaller than $(\mathbb{A}/\mathbb{B})^{\vee} = N(\mathbb{B})$. Hence the subgroup $(N(\mathbb{B})/\mathbb{D})^{\vee} \subset \mathbb{A}/\mathbb{B}$ acts trivially on the topological lines, and can only act on gapless degrees of freedom coming from the physical boundary.

**gSPTs and igSPTs.** Gapless SPT phases are those in which there is no non-trivial topological line, and hence

$$\mathbb{D} = 1. \tag{64}$$

$\mathbb{B}$ is trivial at low energy, while $\mathbb{A}/\mathbb{B}$ only acts on a gapless sector and is fully conformally broken. Therefore, gSPT phases for a 1-form symmetry in (3+1)d are classified by pairs $(\mathbb{B}, \psi)$, with $\mathbb{B} \subset \mathbb{A}$ and $\psi : \mathbb{B} \to \mathbb{A}^{\vee}$ a homomorphism satisfying the property (60).

This phase is *intrinsically gapless* (igSPT) if and only if there is no Lagrangian algebra of the from $\mathcal{L}_{\mathbb{A},\widehat{\psi}}$ (this ensures that $\mathcal{L}_{\mathbb{A},\widehat{\psi}} \cap \mathcal{L}_{\mathrm{sym}} = 1$) such that $\mathcal{A}_{\mathbb{B},1,\psi} \subset \mathcal{L}_{\mathbb{A},\widehat{\psi}}$. This means that $\psi$ must not admit an extension to $\widehat{\psi} : \mathbb{A} \to \mathbb{A}^{\vee}$ preserving the property (60).

### 3.3 Examples

#### 3.3.1 Cyclic groups $\mathbb{A} = \mathbb{Z}_n$

Any homomorphism between two cyclic groups satisfies $\psi(b)b' = \psi(b')b$. Hence, gSPT phases are classified by a divisor $p|n$ that determines

$$\mathbb{B}_p = \{qx \mid x = 0, \dots, p-1\}, \quad n = pq, \tag{65}$$

and an order $p$ element of $\mathbb{Z}_n^\vee \cong \mathbb{Z}_n$. The latter is of the form $qm$, $m = 0, \dots, p-1$ and determines

$$\psi(qx) = qmx. \tag{66}$$

Any homomorphism $\psi : \mathbb{Z}_q \to \mathbb{Z}_n^\vee$ has a symmetric extension to $\mathbb{Z}_n$, so there are no igSPT phases.

#### 3.3.2 Minimal igSPT $\mathbb{A} = \mathbb{Z}_4 \times \mathbb{Z}_2$

The smallest 1-form symmetry group that admits intrinsically gapless SPT phases is $\mathbb{Z}_4 \times \mathbb{Z}_2$. The subgroup $\mathbb{B} \subset \mathbb{A}$ that is part of the classification data $(\mathbb{B}, \psi)$, here is $\mathbb{B} = \mathbb{Z}_2 = \langle (2,0) \rangle \subset \mathbb{A}$ (for all other subgroups there are no igSPTs). The most general homomorphism $\mathbb{Z}_2 \to \mathbb{Z}_4 \times \mathbb{Z}_2$ is

$$\psi_{s_1, s_2}(2x, 0) = (2s_1 x, s_2 x), \tag{67}$$

with $s_i = 0, 1$. Since $\psi_{s_1, s_2}(2x, 0) \cdot (2x', 0) = 1$, this is automatically symmetric. Moreover, if $s_2 = 1$ it does not admit any extension (neither non-symmetric) $\widehat{\psi} : \mathbb{A} \to \mathbb{A}^\vee$, hence it represents an igSPT phase. We have two of them with $s_1 = 0, 1$. We will come back to this example shortly, providing a concrete model that realizes these phases in the vanilla case of $s_1 = 0$ (turning on $s_1$ corresponds to stacking a standard gapped SPT), whose condensable algebra is

$$\mathcal{A}_{\mathbb{Z}_2, 1, \psi} = \{(2x, 0; 0, x) \mid x = 0, 1\}. \tag{68}$$

The fact that this defines an igSPT can be understood from the reduced topological order, too. Using standard methods one discovers that the reduced theory is the (4+1)d Dijkgraaf-Witten theory for $\mathbb{Z}_4$, whose by electric and magnetic surfaces are generated by:

$$E = (1, 0; 0, 0), \quad M = (0, 1; 1, 0). \tag{69}$$

In order to determine the symmetry $\mathcal{S}'$ we ask which of these surfaces can terminate topologically on $\mathfrak{B}^{\text{sym}}$. The allowed surfaces are:

$$E^2 = (0, 0; 0, 1), \quad M^2 = (0, 0; 2, 0). \tag{70}$$

This boundary condition defines a polarization whose symmetry is $\mathcal{S}' = \mathbb{Z}_2^{(1)} \times \mathbb{Z}_2^{(1)}$ with a mixed anomaly [78]:

$$I = \frac{2\pi i}{2} \int B_1 \cup \beta(B_2), \quad \beta(B_2) = \frac{1}{2}\delta B_2. \tag{71}$$

This proves that the IR symmetry is anomalous, i.e. we are describing an igSPT.

It is worth emphasizing the physical reason why the minimal igSPT for 1-form symmetries is $\mathbb{Z}_2 \times \mathbb{Z}_4$ rather than $\mathbb{Z}_4$ as in (1+1)d. An igSPT is characterized by an emergent anomaly for the faithfully acting symmetry in the IR. While in (1+1)d $\mathbb{Z}_2 = \mathbb{Z}_4 / \mathbb{Z}_2$ can be anomalous since $H^3(\mathbb{Z}_2, U(1)) \cong \mathbb{Z}_2$, in (3+1)d we have $H^5(B^2 \mathbb{Z}_2, U(1)) = 0$, but $H^5(B^2 \mathbb{Z}_2 \times \mathbb{Z}_2, U(1)) \cong \mathbb{Z}_2$. Therefore a UV $\mathbb{Z}_4 \times \mathbb{Z}_2$ symmetry can realize an igSPT if $\mathbb{Z}_2 \subset \mathbb{Z}_4$ trivializes in the IR, and the

quotient $\mathbb{Z}_4 \times \mathbb{Z}_2$ gets an emergent anomaly. Interestingly,[19] however, a $\mathbb{Z}_2$ 1-form symmetry can have a mixed gravitational anomaly that can be detected on non-spin manifold, and has inflow

$$\frac{2\pi i}{2} \int_{X_5} w_2 \cup \beta(B_2), \tag{72}$$

with $w_2$ the second Stiefel-Whitney class of the tangent bundle, $B_2$ the background for the 1-form symmetry $\mathbb{Z}_2$ and $\beta(B_2) = \frac{\delta B_2}{2}$ its Bockstein. While we always work on spin-manifolds in this paper, as a consequence of the previous discussion one can presumably find new igSPT phases that can only be detected by putting the system on non-spin manifolds.

### 3.3.3 Examples: $\mathbb{A} = \mathbb{Z}_n \times \mathbb{Z}_n$

A wider class of examples of gSPT and igSPT phases arises for $\mathbb{A} = \mathbb{Z}_n \times \mathbb{Z}_n$, provided that $n = pq$ can be written as the product of two non coprime integers. Let us present all the details in the $n = 4$ case, and then sketch the generalization to other values of $n$.

The group $\mathbb{Z}_4 \times \mathbb{Z}_4$ has nine non-trivial proper subgroups:

$$\left(\mathbb{Z}_4\right)_L, \left(\mathbb{Z}_4\right)_R, \left(\mathbb{Z}_4\right)_D, \left(\mathbb{Z}_2\right)_L, \left(\mathbb{Z}_2\right)_R, \left(\mathbb{Z}_2\right)_D, \mathbb{Z}_4 \times \mathbb{Z}_2, \mathbb{Z}_2 \times \mathbb{Z}_4, \mathbb{Z}_2 \times \mathbb{Z}_2. \tag{73}$$

For all subgroups isomorphic to a cyclic group, it can be checked that any symmetric homomorphism $\psi : \mathbb{B} \to \mathbb{Z}_4 \times \mathbb{Z}_4$ has a symmetric extension, therefore there are no igSPTs associated with these subgroups.

Let us look at $\mathbb{B} = \mathbb{Z}_2 \times \mathbb{Z}_2$. There are 16 homomorphisms $\mathbb{Z}_2 \times \mathbb{Z}_2 \to \mathbb{Z}_4 \times \mathbb{Z}_4$ that we label with four parameters $s_1, s_2, r_1, r_2 = 0, 1$:

$$\psi_{s_1,s_2,r_1,r_2}(2x, 2y) = (2s_1 x + 2r_1 y, 2s_2 x + 2r_2 y). \tag{74}$$

Notice that $\psi_{s_1,s_2,r_1,r_2}(2x, 2y)(2x', 2y') = 1$, so these are all symmetric and define gSPTs. Each homomorphism $\psi_{s_1,s_2,r_1,r_2}$ has 16 extensions, parametrized by other four labels $\sigma_1, \sigma_2, \rho_1, \rho_2 = 0, 1$:

$$\widehat{\psi}^{\sigma_1,\sigma_2,\rho_1,\rho_2}_{s_1,s_2,r_1,r_2}(x, y) = \Big((s_1 + 2\sigma_1)x + (r_1 + 2\rho_1)y, (s_2 + 2\sigma_2)x + (r_2 + 2\rho_2)y\Big), \tag{75}$$

that is symmetric if and only if $r_1 + 2\rho_1 = s_2 + 2\sigma_2$. We conclude that if $r_1 \neq s_2$ there is no symmetric extension of $\widehat{\psi}_{s_1,s_2,r_1,r_2}$, hence this represents an igSPT phase. We have eight igSPTs of this kind.

The last subgroup to consider is $\mathbb{B} = \mathbb{Z}_4 \times \mathbb{Z}_2$, for which there are no igSPTs. Indeed the most general homomorphism is

$$\widehat{\psi}_{s_1,s_2,r_1,r_2}(x, 2y) = (s_1 x + 2r_1 y, s_2 x + 2r_2 y), \tag{76}$$

where $s_1, s_2 = 0, 1, 2, 3$ while $r_1, r_2 = 0, 1$, and is symmetric if and only if $s_2 \bmod(2) = r_1$. There are four extensions

$$\widehat{\psi}^{\rho_1,\rho_2}_{s_1,s_2,r_1,r_2}(x, y) = \Big(s_1 x + (r_1 + 2\rho_1)y, s_2 x + (r_2 + 2\rho_2)y\Big), \tag{77}$$

for which the symmetric condition is $s_2 = r_1 + 2\rho_1$, that has solution precisely if $s_2 \bmod(2) = r_1$.

The igSPTs of $\mathbb{A} = \mathbb{Z}_4 \times \mathbb{Z}_4$ have a natural generalization for $\mathbb{A} = \mathbb{Z}_n \times \mathbb{Z}_n$. Consider $n = pq$, and we look at the subgroup $\mathbb{B} = \mathbb{Z}_p \times \mathbb{Z}_p$. Symmetric homomorphisms $\mathbb{Z}_p \times \mathbb{Z}_p \to \mathbb{Z}_n \times \mathbb{Z}_n$ are

$$\psi_{s1,s_2,r_1,r_2}(qx, qy) = \Big(qs_1 x + qr_1 y, qs_2 x + qr_2 y\Big), \tag{78}$$

---

[19]We thank an anonymous SciPost referee for pointing this out.

Table 6: 1-Form symmetry groups and background fields.

| Group | $\mathbb{Z}_2^e$ | $\mathbb{Z}_2^m$ | $\mathbb{Z}_2^{e'}$ | $\mathbb{Z}_2^{m\vee}$ | $\mathbb{Z}_2^{e'\vee}$ | $\mathbb{Z}_4$ |
|---|---|---|---|---|---|---|
| Background Field | $B_e$ | $B_m$ | $B_e'$ | $\widehat{B}_m$ | $\widehat{B'}_e$ | $2\widehat{B}_m + B_e$ |

with $s_1, s_2, r_1, r_2 \in \mathbb{Z}_p$ and

$$r_1 = s_2 \bmod \left( \frac{p}{\gcd(p,q)} \right). \tag{79}$$

The difference $r_1 - s_2$ can then take $\gcd(p,q)$ values. Therefore there are $p^3 \gcd(p,q)$ gSPTs. To check which of them are igSPTs we look for the extensions $\widehat{\psi} : \mathbb{A} \to \mathbb{A}^\vee$, that are parametrized by $\sigma_i, \rho_i \in \mathbb{Z}_q$

$$\widehat{\psi}^{\sigma_1,\sigma_2,\rho_1,\rho_2}_{s_1,s_2,r_1,r_2}(x,y) = \Big( (s_1 + p\sigma_1)x + (r_1 + p\rho_1)y, (s_2 + p\sigma_2)x + (r_2 + p\rho_2)y \Big), \tag{80}$$

and for this to be symmetric, i.e. for an extension to exist, the condition is

$$r_1 + p\rho_1 = s_2 + p\sigma_2 \ \bmod(n). \tag{81}$$

Thus if $r_1 = s_2 \bmod p$ there is a symmetric extension. Otherwise, there is no one, and in the latter case we get an igSPT. We conclude that, among the $p^3 \gcd(p,q)$ gSPTs, $p^3$ are non-intrinsic, while the remaining $p^3(\gcd(p,q) - 1)$ are igSPTs.

## 3.4 Physical realization of igSPT phases

We can take as input any of the igSPT phases we found and produce a physical IR theory that realizes it similarly to the (1+1)d construction of [76]. To illustrate the idea, we consider the realization of the minimal example of $\mathbb{A} = \mathbb{Z}_4 \times \mathbb{Z}_2$, which is associated with $\mathbb{B} = \mathbb{Z}_2$ and $\psi(2a,0) = (0,a)$. We take a CFT $\mathfrak{T}_0$ with 1-form symmetry $\mathbb{Z}_4^{\text{in}}$ and construct $\mathfrak{T}_0/\mathbb{Z}_2$ by gauging the $\mathbb{Z}_2 \subset \mathbb{Z}_4^{\text{in}}$ subgroup of the 1-form symmetry. Since $\mathbb{Z}_4^{\text{in}}$ is a non-trivial extension, in the resulting theory the dual $\mathbb{Z}_2^m = \mathbb{Z}_2^\vee$ 1-forms symmetry and the quotient $\mathbb{Z}_2^e = \mathbb{Z}_4^{\text{in}}/\mathbb{Z}_2$ have a mixed 't Hooft anomaly

$$\frac{2\pi i}{2} \int_{X_5} B_e \cup \beta(B_m), \tag{82}$$

with $B_m$ the background for the dual symmetry $\mathbb{Z}_2^m$, and $B_e$ the background for the quotient $\mathbb{Z}_2^e$. We then stack a completely trivial theory with 1-form symmetry $\mathbb{Z}_2^{e'}$, and use $\psi$ to construct an SPT involving $B_m$ and the background $B_e'$ for the decoupled $\mathbb{Z}_2^{e'}$:

$$\frac{2\pi i}{2} \int_{X_4} B_m \cup B_e'. \tag{83}$$

Finally, we gauge $\mathbb{Z}_2^m \times \mathbb{Z}_2^{e'}$ with this SPT. Following similar steps as in section 2.4.1 we find that the resulting theory has symmetry $\mathbb{Z}_4 \times \mathbb{Z}_2$. The $\mathbb{Z}_4$ part arises as a non-trivial extension:

$$1 \to \mathbb{Z}_2^{m\vee} \to \mathbb{Z}_4 \to \mathbb{Z}_2^e \to 1. \tag{84}$$

Denoting by $\widehat{B}_m$ the background field for $\mathbb{Z}_2^{m\vee}$, there is a modified cocycle condition

$$\delta\widehat{B}_m = \beta(B_e). \tag{85}$$

The partition function of the resulting theory, that we denote by $\mathfrak{T}$, is

$$Z_{\mathfrak{T}}\left[\widehat{B}_m, B_e, \widehat{B'}_e\right] = \exp\left(-\frac{2\pi i}{2}\int_{X_4}\widehat{B}_m \cup \widehat{B'}_e\right)Z_{\mathfrak{T}_0/\mathbb{Z}_2}\left[\widehat{B'}_e, B_e\right]. \tag{86}$$

Here $\widehat{B'}_e$ is the background for the dual symmetry $\mathbb{Z}_2^{e'\vee}$ (see table 6 for a summary of the symmetries and background fields involved). As we will see shortly in a concrete model, this piece of the symmetry acts trivially on the dynamical CFT, that is $\mathfrak{T}_0/\mathbb{Z}_2$. The same is clearly true also for the $\mathbb{Z}_2 \subset \mathbb{Z}_4$ subgroup, as its background $\widehat{B}_m$ only appears in the multiplying phase. This is the same result as in (1+1)d: the the anomaly of the dynamical part $\mathfrak{T}_0/\mathbb{Z}_2$ is cancelled by the Green-Schwarz mechanism due to the modified cocycle condition of $\widehat{B}_m$ and the presence of the multiplying phase. Hence the CFT has an emergent anomaly that forbids from trivially gapping it by IR deformations. However, remembering the presence of the symmetry $\mathbb{Z}_2^{m\vee}$ only acting on gapped degrees of freedom, the anomaly is cancelled and it is possible to drive the system to a trivially gapped phase by a UV deformation. This would, however, require to make gapless some of the degrees of freedom on which $\mathbb{Z}_2^{m\vee}$ is acting, hence closing the symmetry gap and encountering a phase transition.

## 3.5 An $SU(4) \times SU(2)$ gauge theory realization of igSPTs

This somewhat formal discussion can be made concrete in a model model where all the above ingredients are embedded into an asymptotically free gauge theory. For instance, the CFT $\mathfrak{T}_0$ with 1-form symmetry $\mathbb{Z}_4^{\text{in}}$ can be realized as the fixed point of a $SU(4)$ gauge theory with enough massless adjoint fermions $\psi$ to land in the conformal window. For the trivial theory with $\mathbb{Z}_2$ symmetry we could simply take pure $SU(2)$ YM theory, but without affecting the IR we can replace it with $SU(2)$ gauge theories with massive adjoint fermions $\chi$.[20] This fact will be relevant at the end of the analysis.

Let us analyze the various line operators, and re-derive the above result in a more physical way, interpreting it in terms of confinement/deconfinement. We denote by $W^a$ and $T^b$, $a, b = 0, \dots, 3$ the Wilson and (non-genuine) 't Hooft lines of the $SU(4)$ sector, while by $W', T'$ the analogous lines in the $SU(2)$ sector. Since the first sector is designed to flow in the conformal window, all the $W^a$ lines have perimeter law, while $T^b$ have area law and disappear from the CFT. Vice versa, $W'$ and $T'$ have, respectively, area and perimeter law. In the original $SU(4) \times SU(2)$ global variants, all the Wilson lines are genuine, while the 't Hooft lines are non-genuine, i.e. live in twisted sectors. In the IR this flows to a CFT with $\mathbb{Z}_4$ 1-form symmetry times a trivially gapped (confined) phase with $\mathbb{Z}_2$ zero-form symmetry. The phase is not protected by symmetry, as turning on a mass for the adjoints $\psi$ leads to a trivially gapped (confined) theory.

We then pass to the $SU(4)/\mathbb{Z}_2$ global variant. Now the 1-form symmetry is

$$\mathbb{Z}_2^e \times \mathbb{Z}_2^m \times \mathbb{Z}_2^{e'}, \tag{87}$$

and the genuine lines are $W^2, T^2$ and $W'$, which are charged respectively under the three $\mathbb{Z}_2$ factors. We then gauge $\mathbb{Z}_2^m \times \mathbb{Z}_2^{e'}$ adding the discrete torsion term (83), whose effect is to change the lines that become genuine. In fact, before promoting $B_m, B'_e$ to dynamical fields, the lines $W(W^3), T, T'$ are in twisted sectors, and the presence of the counterterm (83) changes their charges as in table 7. The factors of $-1$ stem from the stacking with the SPT phase, while the fractionalized charge $i$ describes the mixed 't Hooft anomaly

$$\pi i \int B_e \cup \beta(B_m). \tag{88}$$

---

[20]The reason for this choice is that, with the correct number of $SU(2)$ adjoints, in the UV we may play with their mass, eventually reaching the massless point so that the $SU(2)$ sector also reaches a conformal point.

Table 7: Charges of twisted lines under the various symmetries.

|  | $\mathbb{Z}_2^e$ | $\mathbb{Z}_2^m$ | $\mathbb{Z}_2^{e'}$ | Twisted Sector |
|---|---|---|---|---|
| $T$ | 1 | $i$ | 1 | $\mathbb{Z}_2^e$ |
| $W$ | $i$ | 1 | $-1$ | $\mathbb{Z}_2^m$ |
| $T'$ | 1 | $-1$ | 1 | $\mathbb{Z}_2^{e'}$ |

Table 8: Confinement pattern of both genuine and non-genuine lines in the final theory $\mathfrak{T}$.

| Line | $WW'$ | $W^2$ | $T'T^2$ | $W$ | $W'$ | $T'$ | $T$ |
|---|---|---|---|---|---|---|---|
| Area/Perimeter | $A$ | $P$ | $A$ | $P$ | $A$ | $P$ | $A$ |
| Genuine/Twisted | genuine | genuine | genuine | $\mathbb{Z}_2^{e'\vee}$ | $\mathbb{Z}_2^{e'\vee}$ | $\mathbb{Z}_2^{m\vee}$ | $\mathbb{Z}_4$ |

After gauging $\mathbb{Z}_2^m \times \mathbb{Z}_2^{e'}$, on top of keeping only the invariant lines (only $W^2$ in this case), we have to add the twisted sector lines that are *not* charged under $\mathbb{Z}_2^m \times \mathbb{Z}_2^{e'}$. Hence, the lattice of genuine lines is generated by

$$WW' \quad \text{and} \quad T'T^2 , \tag{89}$$

as opposed to $W$ and $T'$, and the symmetry is $\mathbb{Z}_4 \times \mathbb{Z}_2^{e'\vee}$. The lines $W$ and $W'$ both go into the twisted sector of $\mathbb{Z}_2^{e'\vee}$ (whilst their product $WW'$ is genuine), $T$ goes into the twisted sector of $\mathbb{Z}_4$, while $T'$ into the twisted sector of the subgroup $\mathbb{Z}_2^{m\vee}$ (indeed $T'T^2$ is genuine).

Following our dynamical assumptions, before the discrete manipulation, $W$ and $T'$ are deconfined, while $W'$ and $T$ are confined. Thus in the low-energy theory the only genuine line present is $W^2$, as both $WW'$ and $T'T^2$ have area law. This indicates that only a $\mathbb{Z}_2$ quotient of the 1-form symmetry acts in the IR theory, which does not contain topological lines. Hence, this $\mathbb{Z}_2$ quotient acts on the gapless sector. In the twisted sectors, instead, $T$ and $W'$ have area law, while $T'$ and $W$ have perimeter law. Thus, both $\mathbb{Z}_4^e$ and $\mathbb{Z}_2^{m'}$ can be detected at low energy by looking at the twisted sectors. The reader can consult table 8 for a clear synthesis.

### 3.5.1 Deformation by fermion masses

We can now characterize the peculiar feature of this topological phase in this physical setup. Suppose that we try to deform the IR in order to reach a fully confined phase. Since SSB for 1-form symmetry is detected by reducing on an $S^1$ we can think of a finite-temperature setup where a monopole potential, as well as fermion masses, can be turned on if allowed in the IR. Let us first focus on fermion masses. To completely confine the theory, we should confine the $W^2$ line. This can be done, for example, by turning on an equal mass for all $SU(4)$ adjoints.[21] This would imply that $T$ has perimeter law. Notice, however, that $T$ is no longer present in the low energy description. Furthermore, if we give $T$ a perimeter law, then $T'T^2$ is also deconfined, and we would break $\mathbb{Z}_2^{e'\vee}$ spontaneously.

---

[21]We assume that the fermion mass operator is still relevant in the IR CFT.

At first glance, this may seem akin to a (3+1)d CFT with an anomalous 1-form symmetry, which prevents the theory from being trivially gapped. However, the unique aspect of this phase is the absence of an anomaly for the full UV symmetry $\mathcal{S}$. This indicates that the theory might become fully confined if we can render some gapped degrees of freedom massless initially. For example, by tuning to zero the UV mass for the $SU(2)$ adjoints we can drive this sector to conformality in the IR.

Consequently, the $WW'$ line has perimeter law, and the full $\mathbb{Z}_4$ acts non-trivially on the gapless degrees of freedom. This represents a gapless SPT phase, but it is the non-intrinsic one. In fact, it can be deformed into the trivial phase by uniformly increasing the masses for the low-energy adjoints of both $SU(4)$ and $SU(2)$.

However, this deformation is not possible in the igSPT phase since, there, the $SU(2)$ adjoints have already been integrated out, and their mass operator is not part of the CFT. Furthermore, to make this deformation possible, certain degrees of freedom need to become massless, leading to a phase transition, supporting the assertion that the igSPT we are discussing is a distinct phase.

### 3.5.2 Deformation by monopole potentials

Finally, let us comment on robustness of this phase against another type of deformation, which is important to distinguish this phase from more familiar ones. The fact that giving a mass to the $SU(4)$ adjoint fermions drives the CFT into a $\mathbb{Z}_2$ SSB phase should not be surprising, and we would have found a similar result starting from $PSU(4)$ gauge theory instead. However, in the latter, the SSB phase is not protected by any mechanism.

In fact, the situation is different once we study the deformations obtained by reducing on $S^1$ and turning on monopole potentials. Here we use the abuse of terminology by which any local operator obtained wrapping a line operator on $S^1$ is called a monopole, independently of whether it is charged under an electric or a magnetic symmetry.

In the $PSU(4)$ theory on $S^1 \times \mathbb{R}^3$ we can turn on a monopole potential

$$V(\mathcal{W}\mathcal{W}^\dagger), \tag{90}$$

where $\mathcal{W}$ is the Polyakov loop. By carefully choosing the potential, this condenses $\mathcal{W}$ and leads to the $PSU(4)$ confined phase. Notice that we can turn on this potential in the IR since $W$ has a perimeter law.

In the igSPT case, instead, we would like to turn on a potential for $\mathcal{M}\mathcal{M}^\dagger$:

$$V(\mathcal{M}\mathcal{M}^\dagger), \tag{91}$$

where $\mathcal{M}$ is the reduction of the 't Hooft line on $S^1$. However, table 8 shows that $T$ has area law, thus $\mathcal{M}$ is absent in the IR and this deformation is not available in the low-energy theory. We can only turn on a potential for $\mathcal{W}^2(\mathcal{W}^2)^\dagger$, which would lead us too to a SSB phase for $\mathbb{Z}_4^e$. Thus, the igSPT phase cannot be trivially gapped by IR deformations.

## 4 Gapless phases with duality symmetries

We now turn to exploring (intrinsically) gapless SPT phases for selected non-invertible symmetries. In this section we will focus on duality-type symmetries in both (1+1)d and (3+1)d, giving a SymTFT construction and discussing the constraints on the IR physics from the perspective of the gapless sector.[22]

---

[22]The methods and classification developed here generalize in a straightforward manner to $G$-ality defects. See [75, 84, 85] for recent studies.

## 4.1 Phases with duality symmetries: (1+1)d

We want to generalize the analysis of the gapless phases in (1+1)d to a larger class of fusion categories $\mathcal{S}$ whose Drinfeld center $\mathcal{Z}(\mathcal{S})$ is obtained from some abelian topological order $\mathcal{Z}(\text{Vec}_{\mathbb{A}})$ by gauging a finite invertible 0-form symmetry $G$.[23] Examples are $\text{Vec}_{D_{2n}}$, whose Drinfeld center is obtained by gauging charge conjugation in $\mathcal{Z}(\text{Vec}_{\mathbb{Z}_n})$, and Tambara-Yamagami categories $\text{TY}(\mathbb{A}, \gamma, \epsilon)$ [7,75,86,87], whose center is obtained from $\mathcal{Z}(\text{Vec}_{\mathbb{A}})$ gauging electro-magnetic duality [38,41,50,88]. Later we will extend our approach to $(3+1)$d. Although the center of TY-categories is well-known [88], our approach gives a construction of the center which extends more easily to higher dimensions (see also [41,50]). Our goal will be to determine the condensable algebras, including SPT, gSPT, and most interestingly igSPTs, through the gauging.

### 4.1.1 Structure of the center

We consider a faithful 0-form symmetry group $G$ acting on $\mathbb{A} \times \mathbb{A}^\vee$ by exchanging anyons, while preserving fusion and braiding. Practically, there exists a group homomorphism $\Phi : G \to \text{Aut}(\mathbb{A} \times \mathbb{A}^\vee)$ such that

$$\theta_{\Phi_g(a,\alpha)} = \theta_{(a,\alpha)}, \qquad \forall a \in \mathbb{A}, \alpha \in \mathbb{A}^\vee. \tag{92}$$

Faithfulness means that $\Phi$ defines an embedding $G \subset \text{Aut}_0(\mathbb{A} \times \mathbb{A}^\vee)$ in the subgroup $\text{Aut}_0(\mathbb{A} \times \mathbb{A}^\vee) \subset \text{Aut}(\mathbb{A} \times \mathbb{A}^\vee)$ that preserves the pairing $(a, \alpha) \mapsto \alpha(a) \in U(1)$.

**Structure of $\text{Aut}_0(\mathbb{A} \times \mathbb{A}^\vee)$.** $\text{Aut}_0(\mathbb{A} \times \mathbb{A}^\vee)$ is generated by three subgroups [89]: one is isomorphic to $\text{Aut}(\mathbb{A})$, another to $H^2(\mathbb{A}, U(1))$, while the third to $\mathbb{Z}_2$. The embeddings $\text{Aut}(\mathbb{A})$ and $H^2(\mathbb{A}, U(1))$ into $\text{Aut}_0(\mathbb{A} \times \mathbb{A}^\vee)$ are canonical. Given an automorphism $\rho : \mathbb{A} \to \mathbb{A}$, we have $\rho^{-1\vee} : \mathbb{A}^\vee \to \mathbb{A}^\vee$ given by $\rho^{-1\vee}(\alpha)(a) = \alpha(\rho^{-1}(a))$. Hence $P_\rho \in \text{Aut}_0(\mathbb{A} \times \mathbb{A}^\vee)$ is given by

$$P_\rho(a, \alpha) = \left(\rho(a), \rho^{-1\vee}(\alpha)\right). \tag{93}$$

Similarly, given $\omega \in H^2(\mathbb{A}, U(1))$, $\chi : \mathbb{A} \times \mathbb{A} \to U(1)$ be the associate alternating bicharacter and $\psi : \mathbb{A} \to \mathbb{A}^\vee$ the corresponding group homomorphism, $Q_\omega \in \text{Aut}_0(\mathbb{A} \times \mathbb{A}^\vee)$ is

$$Q_\omega(a, \alpha) = (a, \psi(a)\alpha). \tag{94}$$

The $\mathbb{Z}_2$ subgroup, instead, is generated by electro-magnetic duality $S$ and its identification is non-canonical. Given a *choice* of isomorphism $\phi : \mathbb{A} \to \mathbb{A}^\vee$ such that $\phi^\vee = \phi$ — namely $\phi(a)b = \gamma(a, b)$ is a symmetric bicharacter — then $S$ is defined as[24]

$$S(a, \alpha) = \left(\phi^{-1}(\alpha), \phi(a)\right), \tag{95}$$

and preserves the spins because of $\phi(a)b = \phi(b)a$.

In [89] it was proved that these three subgroups exhausted all the elements of $\text{Aut}_0(\mathbb{A} \times \mathbb{A}^\vee)$. One physical interpretation is obtained by fixing the starting symmetry boundary to be the canonical Dirichlet one realizing the symmetry $\text{Vec}_{\mathbb{A}}$, and each 0-form symmetry of the SymTFT is a surface that maps this symmetry boundary to a different one. Then the elements of $\text{Aut}(\mathbb{A})$ simply permute the symmetry generators, leaving that boundary invariant up to isomorphism, those of $H^2(\mathbb{A}, U(1))$ maps the boundary adding SPT phases, while $S$ maps it to its electro-magnetic dual.

---

[23]Without loss of generality we can assume $G$ to act faithfully on $\mathcal{Z}(\text{Vec}_{\mathbb{A}})$.

[24]It is easy to check that two different choices $\phi_1, \phi_2$ lead to $S_2, S_1$ such that $S_2 = P_\rho S_1$, with $\rho = \phi_2^{-1}\phi_1 \in \text{Aut}(\mathbb{A})$.

Table 9: Objects (lines) of the 3d Symmetry TFT for the Tambara-Yamagami symmetry.

| Object | Definition | Dim | # of Objects | Spin $\theta$ |
|--------|-----------|-----|--------------|---------------|
| $L_{(a,x)}$ | $\eta^x \times (a, \phi(a))$ | 1 | $2\lvert\mathbb{A}\rvert$ | $\gamma(a,a)$ |
| $X_{(a,b)}$ | $(a, \phi(b)) \oplus (b, \phi(a))$ | 2 | $\lvert\mathbb{A}\rvert(\lvert\mathbb{A}\rvert-1)/2$ | $\gamma(a,b)$ |
| $\Sigma_{(a,x)}$ | $\eta^x \times \sigma_a$ | $\sqrt{\lvert\mathbb{A}\rvert}$ | $2\lvert\mathbb{A}\rvert$ | $(-1)^x \sqrt{\dfrac{\epsilon}{\lvert\mathbb{A}\rvert^{1/2}} \sum_{b\in\mathbb{A}} \overline{f_a(b)^{-1}}}$ |

**Anyons of the gauged center.** We consider the fusion categories $\mathcal{S}$ such that $\mathcal{Z}(\mathcal{S}) = \mathcal{Z}(\mathsf{Vec}_{\mathbb{A}})/G$. To fix our notation and language, let us briefly review the anyon content of $\mathcal{Z}(\mathcal{S})$, in a way that can be generalized to higher dimensions [41, 42, 49, 50]. As this is a well-known procedure, we refer the reader to [90] for details. In the following, we only consider the case of $G$ abelian and, in particular, $G = \mathbb{Z}_2$. Anyons in $\mathcal{Z}(\mathcal{S})$ fall into three classes:

- $G$-invariant combinations (orbits) of anyons of $\mathcal{Z}(\mathsf{Vec}_{\mathbb{A}})$. For $G = \mathbb{Z}_2$ they can be long orbits $X_{a,b}$ or invariant lines $L_{a,\pm}$.

- Lines $\eta_r$, $r \in \mathsf{Rep}(G) \cong G^\vee$ of the dual symmetry.

- Twist defects $\Sigma_{(a,x)}$, coming from the $G$-twisted sectors $\sigma_a$ prior to the gauging. For $G = \mathbb{Z}_2$, we have $x = \pm$. These are the charged objects under the $\mathsf{Rep}(G)$ symmetry.

The suffix $\pm$ indicates that the $\mathsf{Rep}(G)$ symmetry line can fuse with the defect giving rise to a new topological object. We summarize their structure, in the case of Tambara-Yamagami categories, in table 9. The case of $\mathsf{Vec}_{D_{4n}}$ appeared in [38] and is instead reviewed in appendix A. The reader is referred to [41, 88] for details of the complete categorical structure.

### 4.1.2 Condensable algebras in $\mathcal{Z}(\mathcal{S})$

In the present context, useful information about $\mathcal{Z}(\mathcal{S})$ can be inferred from the fact that $\mathcal{Z}(\mathsf{Vec}_{\mathbb{A}})$ and $\mathcal{Z}(\mathcal{S})$ are connected to each other by gauging $G$ or $\mathsf{Rep}(G)^{(1)} = G^\vee$. To start with, we consider the open club sandwiches, defined by condensable algebras. These define a topological interface[25]

$$
\begin{array}{c}
\mathcal{I}_{\mathsf{Rep}(G)} \\[4pt]
\boxed{\quad \mathcal{Z}(\mathsf{Vec}_{\mathbb{A}}) \;\Big|\; \begin{array}{c} \mathcal{Z}(\mathsf{Vec}_{\mathbb{A}})/G \\ = \mathcal{Z}(\mathcal{S}) \end{array} \quad} \\[4pt]
\text{G-gauging interface}
\end{array}
\tag{96}
$$

---

[25]The study of such interfaces has a long history, dating back to [91–93], see also [19, 94] for recent studies in $(1+1)$d.

The topological interface is a G-gauging interface, defined by gauging $G$ in the right-half of the space with Dirichlet boundary conditions for the $G$ gauge field. Alternatively, it is associated with the condensation of the algebra

$$\mathcal{A}_G = \bigoplus_{r \in \text{Rep}(G)} \eta_r \,, \tag{97}$$

of $\mathcal{Z}(\mathcal{S})$. From the point of view of $\mathcal{Z}(\mathcal{S})$, $\mathcal{Z}(\text{Vec}_{\mathbb{A}})$ can be understood as a reduced topological order. We can construct a lift from *any* condensable algebra $\mathcal{A}_0$ of $\mathcal{Z}(\text{Vec}_{\mathbb{A}})$ to a condensable algebra $\mathcal{A}_0^G$ of $\mathcal{Z}(\mathcal{S})$. Any condensable algebra $\mathcal{A}_0$ defines an interface $\mathcal{I}_{\mathcal{A}_0}$ (and viceversa), and $\mathcal{A}_0^G$ is the algebra corresponding to the stacking of the two interfaces

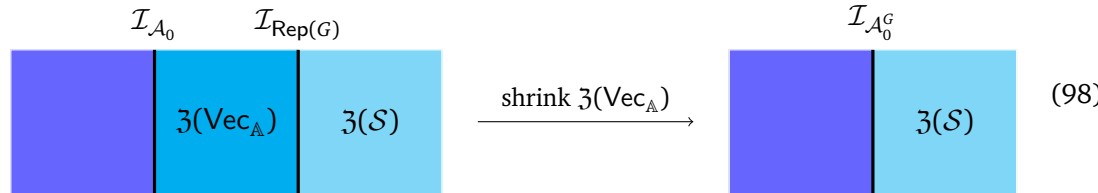

$$\tag{98}$$

We call these algebras *induced algebras* from $\mathcal{A}_0$. These do not exhaust all the possible condensable algebras in $\mathcal{Z}(\mathcal{S})$ as they will all condense the Rep($G$) lines. As appreciated in [50], the missing algebras are lifts of $G$-invariant algebras $\mathcal{A}_I$ in $\mathcal{Z}(\text{Vec}_{\mathbb{A}})$:[26]

$$\Phi(\mathcal{A}_I) = \mathcal{A}_I \,. \tag{99}$$

Having found an invariant algebra, we must also specify a way in which the symmetry $G$ acts on the algebra structure of $\mathcal{A}_I$. This defines an equivariantization of $\mathcal{A}_I^\eta$ of $\mathcal{A}_I$, and can be characterized precisely (see [50] Sec. (3.4)). We will often omit this datum unless relevant for the phase described. An intuitive justification of this construction follows from considering the reduced topological order $\mathcal{Z}(\text{Vec}_{\mathbb{A}})/\mathcal{A}_I$ and gauging the $G$ symmetry:

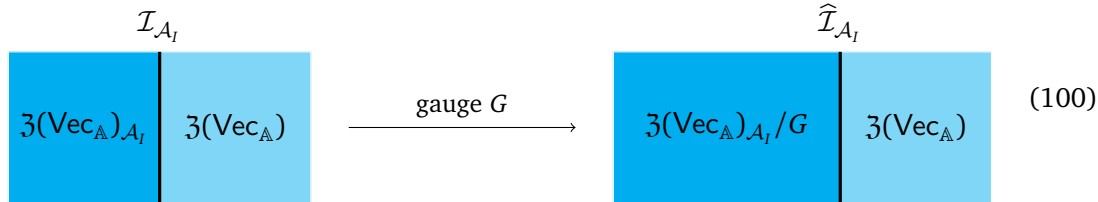

$$\tag{100}$$

This setup is transparent to the Rep($G$) lines and thus defines a new type of interface, denoted by $\widehat{\mathcal{I}}_{\mathcal{A}_I}$. This describes the most general reduced topological order in which the $G$ symmetry is still present. It is possible to find physical examples in which the $G$ symmetry does not act faithfully on gapless degrees of freedom without being broken. We will not discuss these examples, but let us briefly give a flavor of how they are achieved.

Consider a lift of an invariant algebra $\mathcal{A}_I$. Since it is duality-invariant, we can often enlarge it by including twisted sector operators $\Sigma_{a,x}$. This eliminates the $G$ symmetry in the IR description by gauging it. Alternatively, if invariant anyons $L_a$ are present in $\mathcal{A}_I$, and we call $\mathbb{B}_{inv}$ the subgroup that they form, we can decorate $\widehat{\mathcal{A}}_I$ with dual symmetry lines $\eta_r$ using a group homomorphism $\xi : \mathbb{B}_{inv} \to G$. This eliminates the $G$ symmetry from the IR description as it is nonlocal with respect to a nontrivial decoration.

---

[26]Strictly speaking, preserving a subgroup of $G$ is sufficient, but our examples will all be for $G = \mathbb{Z}_2$.

### 4.1.3 gSPT and igSPT phases for $\mathcal{S}$

The structure we have just introduced allows for an efficient description of (i)gSPT phases for $\mathcal{S}$. First of all, notice that the canonical Lagrangian algebra $\mathcal{L}_{\text{symm}}$ that fixes the symmetry to be $\mathcal{S}$ is nothing but $\mathcal{L}_0^G$, induced by

$$\mathcal{L}_0 = \left\{ (0, \alpha) | \, \alpha \in \mathbb{A}^\vee \right\}, \tag{101}$$

which in turn is the canonical Lagrangian algebra of $\mathcal{Z}(\text{Vec}_\mathbb{A})$ that gives the symmetry $\text{Vec}_\mathbb{A}$. As $\mathcal{L}_{\text{symm}}$ contains all the $\text{Rep}(G)$ lines, a necessary condition for a condensable algebra $\mathcal{A}$ to correspond to a gSPT phase is:

$$\mathcal{A} \cap \text{Rep}(G) = \{1\}. \tag{102}$$

Thus $\mathcal{A}$ must be a lift of an invariant algebra $\mathcal{A}_I$. Clearly $\mathcal{A}_I$ is not just *any* invariant algebra in $\mathcal{Z}(\text{Vec}_\mathbb{A})$, but must be a gSPT algebra in order to avoid an SSB phase of $\mathbb{A}$.

However, what we are really interested in is determining which of these gapless SPT phases are intrinsic. We then have to ask if a gSPT algebra $\widehat{\mathcal{A}_I}$ is a subalgebra of a Lagrangian algebra $\mathcal{L}$ corresponding to a (gapped) SPT. We will specialize to the case in which the duality symmetry acts faithfully on gapless degrees of freedom. The general case is also treatable through our formalism, but the complete classification becomes cumbersome. To continue the discussion, we use the result [49, 50] that (gapped) SPTs of $\mathcal{Z}(\mathcal{S})$ can only arise from $G$-invariant Lagrangian algebras $\mathcal{L}_I$ of $\mathcal{Z}(\text{Vec}_\mathbb{A})$. The corresponding SPT algebra $\widetilde{\mathcal{L}_I}$ of $\mathcal{Z}(\mathcal{S})$ is obtained as follows. First, we construct $\widehat{\mathcal{L}_I}$ that is condensable but non-maximal in $\mathcal{Z}(\mathcal{S})$. The reduced topological order is a Dijkgraaf-Witten theory $\mathcal{Z}(\text{Vec}_G^\omega)$. If the cocycle $\omega \in H^3(G, U(1))$ is trivial[27] one can further condense the magnetic lines of $\mathfrak{Z}(\text{Vec}_G)$. This sequence of condensations defines a gapped boundary of $\mathfrak{Z}(\mathcal{S})$, and $\widetilde{\mathcal{L}_I}$ is the corresponding Lagrangian algebra.

Using these facts it is easy to see that if $\mathcal{A}_I$ is a $G$-invariant igSPT algebra of $\mathcal{Z}(\text{Vec}_\mathbb{A})$, then $\widehat{\mathcal{A}_I}$ is also intrinsic.[28] Interestingly, the vice-versa is not necessarily true. We may have a $G$-invariant gSPT algebra $\mathcal{A}_I$ of $\mathcal{Z}(\text{Vec}_\mathbb{A})$ that is non-intrinsic –namely $\mathcal{A}_I \subset \mathcal{L}_0$ for some SPT Lagrangian algebra $\mathcal{L}_0$– but the latter is not $G$-invariant and hence it does not give rise to a gapped SPT of $\mathcal{Z}(\mathcal{S})$. Finally, we can have a Lagrangian algebra $\mathcal{L}_I^\eta$ with non-trivial equivariantization datum $\eta$, whose reduced topological order describes a phase with $\text{Vec}_{\mathbb{Z}_2}^\omega$ symmetry with nontrivial anomaly $\omega$.

To summarize, we can distinguish three types of igSPT phases for theories with $\mathcal{S}$ symmetry:

- **Type I:** igSPT phases of $\text{Vec}_\mathbb{A}$ whose condensable algebra is $G$-invariant.

- **Type II:** gSPT phases of $\text{Vec}_\mathbb{A}$ which are *not* intrisic and whose condensable algebra is $G$-invariant, but such that none of the SPT Lagrangian algebras containing it is $G$-invariant.

- **Type III:** SPT phases for the $\mathbb{A}$ symmetry which are described by a duality invariant Lagrangian algebra $\mathcal{L}_I$, but with nontrivial choice of equivariantization $\mathcal{L}_I^\eta$, which makes them igSPT phases when taking into account duality.

The physical interpretation of Type I and Type II igSPT phases is quite different. Consider, for instance, the case of duality defects, in which $\mathcal{S}$ extends $\text{Vec}_\mathbb{A}$ by adding a non-invertible defect $\mathcal{N}$. A Type I igSPT is an intrinsically gapless phase if we forget the duality, that simply remains so when we remember it, but the presence of $\mathcal{N}$ does not play any fundamental role: the prize to pay for gapping the theory is spontaneously breaking $\mathbb{A}$. On the other hand, a Type II igSPT is such that it is non-intrinsic if we forget duality, hence it can be deformed to a

---

[27] $\omega$ depends not only on the symmetry and the algebra $\mathcal{L}_I$, but also on its equivariantization $\mathcal{L}_I^\eta$.

[28] If it was not, then $\widehat{\mathcal{A}_I} \subset \widetilde{\mathcal{L}_I}$ for some SPT algebra of $\mathcal{Z}(\mathcal{S})$, but then $\mathcal{A}_I \subset \mathcal{L}_I$.

gapped phase if we discard the non-invertible symmetry, but it is intrinsic precisely because of its presence. Hence it is a topological phase protected by the non-invertible symmetry, and if we want to gap the theory, we need to spontaneously break it. Type III phases are similar to Type II phases, in which the full $\mathbb{A}$ symmetry has been realized in a trivially gapped fashion. From the perspective of the gapless degrees of freedom, the duality symmetry is realized in very different manners:[29]

- In **Type I** igSPTs the duality defect is realized as an *invertible* and *anomaly-free* symmetry.

- In **Type II** igSPTs, it is instead realized as a *non-invertible*, but *anomalous* symmetry.

- In **Type III** igSPTs the duality symmetry in the IR is *invertible*, but anomalous.

At this point, it is useful to discuss some concrete example. It turns out that the natural igSPT phases in (1+1)d are of Type I or III, while we will see examples of Type II igSPTs in (3+1)d.

**Type I igSPT phases for $\mathsf{Vec}_{D_8}$.** The Drinfeld center of $\mathsf{Vec}_{D_8}$ is obtained from $\mathcal{Z}(\mathsf{Vec}_{\mathbb{Z}_4})$ by gauging charge conjugation $C : (a, b) \mapsto (-a, -b)$. We refer the reader to appendix A for details and notation.

The group-symmetry $\mathsf{Vec}_{D_8}$ admits two igSPT phases of Type I. Indeed the well known $\mathsf{Vec}_{\mathbb{Z}_4}$ igSPT with algebra

$$\mathcal{A}_{\mathbb{Z}_2, 1, \psi} = \{(2x, 2x) \,|\, x = 0, 1\} \,, \tag{103}$$

is $C$-invariant. Moreover since the line $(2, 2)$ is by itself $C$-invariant, this algebra gives rise to two decorated igSPT algebras in $\mathcal{Z}(\mathsf{Vec}_{D_8})$ in

$$\widehat{\mathcal{A}}^{\pm}_{\mathbb{Z}_2, 1, \psi} = L_{0,0} + L^{\pm}_{2,2} \,. \tag{104}$$

Finally, since the only SPT Lagrangian algebra of $\mathcal{Z}(\mathsf{Vec}_{\mathbb{Z}_4})$, namely $\mathcal{L}_{\mathrm{SPT}} = \{(a, 0)\}$, is $C$-invariant, it gives rise to an SPT for $\mathsf{Vec}_{D_8}$ and there cannot be Type II igSPT phases.

This result matches the finding of [38] by direct analysis of the modular data of the Drinfeld center and the structure of its Hasse diagram. We can also use the present formalism to generalize this example to find igSPT phases for all $\mathsf{Vec}_{D_{2n}}$.

**Type I igSPT phases in $\mathsf{TY}$ categories.** Tambara-Yamagami categories $\mathsf{TY}(\mathbb{A}, \gamma, \epsilon)$ are classified by a finite Abelian group $\mathbb{A}$, a symmetric bicharacter $\gamma : \mathbb{A} \times \mathbb{A} \to U(1)$ and a Froboenius-Schur indicator $\epsilon = \pm 1$ [86]. These data appear naturally in the Drinfeld center [49,50], which can be obtained from $\mathcal{Z}(\mathsf{Vec}_{\mathbb{A}})$ by gauging the electro-magnetic duality (95) determined by a symmetric isomorphism $\phi : \mathbb{A} \to \mathbb{A}^{\vee}$ ($\gamma(a, b) = \phi(a)b$ is the associated bicharacter) with discrete torsion $\epsilon \in H^3(\mathbb{Z}_2, U(1)) = \mathbb{Z}_2$.

Consider the case $\mathbb{A} = \mathbb{Z}_n \times \mathbb{Z}_n$ with off-diagonal bicharacter $\phi_O(x, y) = (y, x)$ and trivial FS indicator, which always admits gapped SPT phases [50]. As we have seen, if $n = pq$, $\gcd(p, q) \neq 1$ there is a class of igSPT for $\mathsf{Vec}_{\mathbb{A}}$, $\mathcal{A}_{\mathbb{Z}_p \times \mathbb{Z}_p, 1, \psi}$ with $\psi$ given by (34). These algebras produce igSPT phases for $TY(\mathbb{Z}_n \times \mathbb{Z}_n, \gamma_O, +)$ if they are duality invariant, and a necessary condition for this to happen is that the image of $\psi$ must be $\mathbb{Z}_p \times \mathbb{Z}_p$. Setting $r = 0$ ($r \neq 0$ corresponds to stacking a gapped SPT) this requires $p = q$: otherwise $t = p/\gcd(p, q)$ is never invertible over $\mathbb{Z}_p$, and the image will be $t\mathbb{Z}_p \times t\mathbb{Z}_p \subset \mathbb{Z}_p \times \mathbb{Z}_p$, which is a proper subset.

---

[29]For readers familiar with the obstruction theory describing anomalies of Tambara-Yamagami categories [7,50,95], Type I igSPTs realize anomalous 0-form symmetries in the IR, Type II igSPTs realize anomalous Tamabara-Yamagami type categories with a nontrivial first obstruction, and Type III igSPTs realize anomalous Tambara-Yamagami categories with trivial first obstruction and nontrivial second obstuction.

The simplest example is $n = 9$, $p = q = 3$, and we choose $k_1 = k_3 = r = 0$, $k_2 = 1$. This gives the algebra

$$\mathcal{A}_{\mathbb{Z}_3 \times \mathbb{Z}_3, 1, \psi} = \{(3x, 3y; 3y, 3x) \mid x, y = 0, 1, 2\}, \tag{105}$$

that is duality invariant for the off-diagonal bicharacter and produces an igSPT phase of of Type I with algebra

$$\widehat{\mathcal{A}}_{\mathbb{Z}_3 \times \mathbb{Z}_3, 1, \psi} = \bigoplus_{x,y=0}^{2} (3x, 3y; 3y, 3x). \tag{106}$$

Notice that, even though all 9 anyons appearing are individually duality invariant, no decoration of this algebra is possible. Indeed, all non-trivial anyons are of order three, hence dressing one of them with the non-trivial generator of the quantum symmetry $\mathrm{Rep}(\mathbb{Z}_2)$ would imply the presence of the dressed identity line, that is not consistent with an SPT.[30]

We also comment on the symmetry realization on the gapless sector. After condensing $\mathcal{A}_{\mathbb{Z}_3 \times \mathbb{Z}_3, 1, \psi}$, the resulting theory describes $\mathcal{Z}(\mathrm{Vec}_{\mathbb{Z}_9})$, with electric and magnetic lines generated by $E = (1, 0; 0, -1)$ and $M = (0, 1; 2, 0)$, respectively. A simple computation shows that our UV choice for duality symmetry descends to charge conjugation $C$. Furthermore, the gapped boundary condition $\mathfrak{B}^{\mathrm{sym}}$ is mapped to the Lagrangian algebra $\mathcal{L}_{\mathbb{Z}_3, 1}$, which describes a theory with $\mathbb{Z}_3 \times \mathbb{Z}_3$ symmetry - with a mixed anomaly - stacked to an invertible $\mathbb{Z}_2$ duality symmetry [50].

**Type III igSPTs in TY categories.** An example of Type III igSPT can also be found in the TY category. Consider $\mathrm{Rep}(D_8) = \mathrm{TY}(\mathbb{Z}_2 \times \mathbb{Z}_2, \gamma_O, +)$. This was the first non-invertible igSPT discovered [38]. We will show that the present formalism reproduces it correctly. This symmetry allows for a duality-invariant SPT by using the Lagrangian algebra:

$$\mathcal{L} = \{(x, y; x, y), \quad x, y = 0, 1\}. \tag{108}$$

The duality symmetry can act on this algebra in multiple manners, called different *equivariantizations* [50]. In practice they are in one-to-one correspondence with one cochains $\eta$ which form a torsor over:

$$H^1_\sigma(\mathbb{Z}_2, \mathcal{L}^\vee), \tag{109}$$

satisfying

$$d_\sigma \eta(b, b') = \left[ \frac{\chi[\psi](b, b')}{\chi[\psi](\sigma(b'), \sigma(b))} \right] = \frac{\gamma_O(b, b')}{\gamma_O(\sigma(b'), \sigma(b))}. \tag{110}$$

Importantly, they induce a 't Hooft anomaly $\omega \in H^3(\mathbb{Z}_2, U(1))$ for the (invertible) duality symmetry after gauging, given by:

$$\omega = \mathrm{Arf}(\eta). \tag{111}$$

In our example $\sigma = 1$ and the r.h.s. of (110) vanishes. Since

$$H^1(\mathbb{Z}_2, \mathbb{Z}_2 \times \mathbb{Z}_2) = \mathrm{Hom}(\mathbb{Z}_2, \mathbb{Z}_2 \times \mathbb{Z}_2) = \mathbb{Z}_2 \times \mathbb{Z}_2,$$

there are four inequivalent choices. In [7,50] it is shown that the $\eta$ map sending the generator of $\mathbb{Z}_2$ to $(1, 1) \in \mathbb{Z}_2 \times \mathbb{Z}_2$ has negative Arf invariant. This describes an igSPT between $\mathrm{Rep}(D_8)$ and $\mathbb{Z}_2^\omega$, as already shown in [38].

---

[30]An example where decoration is possible is $n = 16$, $p = q = 4$, and again $k_1 = k_3 = r = 0$, $k_2 = 1$:

$$\mathcal{A}_{\mathbb{Z}_4 \times \mathbb{Z}_4, 1, \psi} = \{(4x, 4y; 4y, 4x) \mid x, y = 0, \dots, 3\}. \tag{107}$$

This is essentially the same as before, but since as a group $\mathcal{A}_{\mathbb{Z}_4 \times \mathbb{Z}_4, 1, \psi} \cong \mathbb{Z}_4 \times \mathbb{Z}_4$, we can freely assign a representation to each of the two generators of $\mathbb{Z}_4 \times \mathbb{Z}_4$, and this algebra produces $2^2 = 4$ igSPT phases of Type I.

Table 10: First few values of $n$ for which the equation $r^2 = -1 \mod(n)$ has solutions.

| $n$ | 2 | 5 | 10 | 13 | 17 | 25 | 26 | ... |
|---|---|---|---|---|---|---|---|---|
| $r$ | 1 | 2, 3 | 3, 7 | 5, 8 | 4, 13 | 7, 18 | 5, 21 | ... |

## 4.2 Phases with duality symmetries: (3+1)d

The formalism developed in the last subsection can be extended to discuss duality defects in (3+1)d, that arise when a theory is self-dual under gauging a 1-form symmetry $\mathbb{A}^{(1)}$ [1, 2]. We briefly recall how to obtain the Drinfeld center by gauging electro-magnetic duality of $\mathfrak{Z}(\mathbb{A}^{(1)})$ [41, 42], and how to use it to characterized (gapped) SPT phases [49, 50]. We then use the ideas of the last subsection to characterize intrinsically gapless SPT phases, and discuss examples.[31]

### 4.2.1 SPTs for duality defects

**Self-duality symmetry.** As for 0-form symmetries in (1+1)d, also the SymTFT $\mathfrak{Z}(\mathbb{A}^{(1)})$ for 1-form symmetries in (3+1)d has a universal electro-magnetic symmetry determined by a symmetry isomorphism $\phi : \mathbb{A} \to \mathbb{A}^\vee$. The only difference is that, since the braiding in five dimensions (46) is antisymmetric, the duality symmetry is

$$G = \langle S \rangle : (a, \alpha) \mapsto \left(-\phi^{-1}(\alpha), \phi(a)\right), \tag{112}$$

and has order 4. By gauging it, eventually with discrete torsion $\epsilon \in H^5(\mathbb{Z}_4, U(1)) \cong \mathbb{Z}_4$, we get the SymTFT for theories self-dual under gauging $\mathbb{A}^{(1)}$, which include a non-invertible symmetry defect $\mathcal{N}$ [41, 42, 49, 50].

Let us set $\epsilon = 0$ for simplicity. As shown in [49, 50] the SPT phases for the self-duality symmetries are given by $G$-invariant SPT phases for the 1-form symmetry $\mathbb{A}^{(1)}$. These are Lagrangian algebras $\mathcal{L}_{\mathbb{A},\psi}$ such that $S\left(\mathcal{L}_{\mathbb{A},\psi}\right) = \mathcal{L}_{\mathbb{A},\psi}$. This last condition can be translated into the requirements that [50] $\psi : \mathbb{A} \to \mathbb{A}^\vee$ is an isomorphism, and $\sigma := \phi^{-1} \circ \psi$ is a square root of the inversion:

$$\sigma^2 = -1. \tag{113}$$

For $\mathbb{A} = \mathbb{Z}_n$, SPT phases for the 1-form symmetry are determined by $r \in \mathbb{Z}_n$ as

$$\mathcal{L}_r = \{(a, ra) \mid a = 0, \ldots, n-1\}. \tag{114}$$

There is no loss of generality in taking the isomorphism $\phi : \mathbb{Z}_n \to \mathbb{Z}_n^\vee \cong \mathbb{Z}_n$ to be the identity. The two conditions above are that $r \in \mathbb{Z}_n$ must also be invertible, and

$$r^2 = -1 \mod(n), \tag{115}$$

namely $r$ is a quadratic residue of $-1$. In table 10 we report the first few values of $n$ for which this exists.

Since it will be important for us, let us also consider $\mathbb{A} = \mathbb{Z}_n \times \mathbb{Z}_n$. There are two natural choices of isomorphism $\phi$ in the $\mathbb{Z}_n \times \mathbb{Z}_n$ case

- The diagonal $\phi_D(x, y) = (x, y)$.

---

[31]It should be noted that, in higher dimensions, a Lagrangian algebra does not uniquely specify a gapped phase, as decoration by boundary topological order is possible, see e.g. [24, 26, 52, 96]. Our results remain valid, however including these further data will lead to a larger set of igSPTs.

- The off-diagonal $\phi_O(x,y) = (y,x)$.

In the first case a duality invariant SPT is given by a symmetric matrix

$$\psi = \begin{pmatrix} \alpha & \beta \\ \beta & \delta \end{pmatrix}, \tag{116}$$

such that $\psi^2 = -1$. This means that

$$\alpha^2 + \beta^2 = \delta^2 + \beta^2 = -1, \quad (\alpha + \delta)\beta = 0. \tag{117}$$

Taking $\beta = 0$, this again reduces to the existence of a quadratic residue of $-1$. We can also take $\alpha = -\delta$, and the condition becomes[32]

$$\alpha^2 + \beta^2 = -1 \bmod(n). \tag{118}$$

This choice of algebra prescribes perimeter law for the dyonic lines

$$\begin{aligned} W_1^{-\alpha} T_1, \quad W_2^{-\delta} T_2, && \beta = 0, \\ W_1^{-\alpha} W_2^{-\beta} T_1, \quad W_2^{\alpha} W_1^{-\beta} T_2, && \beta^2 = -1 \bmod(n). \end{aligned} \tag{119}$$

On the other hand for $\phi_O$ the existence of duality invariant SPTs is that

$$(\phi_O^{-1}\psi)^2 = \begin{pmatrix} \beta & \delta \\ \alpha & \beta \end{pmatrix}^2 = \begin{pmatrix} \beta^2 + \alpha\delta & 2\delta\beta \\ 2\alpha\beta & \beta^2 + \alpha\delta \end{pmatrix} = -1. \tag{120}$$

This can always be solved by taking $\delta$ to be coprime with $n$ and

$$\alpha = -\delta^{-1}, \quad \beta = 0. \tag{121}$$

Moreover, if there is a quadratic residue of $-1$, it is also solved by $\alpha = \delta = 0$ and :

$$\beta^2 = -1 \bmod(n). \tag{122}$$

Thus, with the off-diagonal isomorphism, the duality defect for a $\mathbb{Z}_n \times \mathbb{Z}_n$ 1-form symmetry has an SPT for any $n$.

The above SPTs are realized by assigning perimeter law to the dyonic lattice generated by:

$$\begin{aligned} W_1^{-\alpha} T_1, \quad W_2^{\alpha^{-1}} T_2, && \beta = 0, \\ W_1^{-\beta} T_2, \quad W_2^{-\beta} T_1, && \beta^2 = -1 \bmod(n). \end{aligned} \tag{123}$$

### 4.2.2 gSPT and igSPT for duality defects

We now look at gapless SPT phases protected by duality symmetry. The general story of sections 4.1.2 and 4.1.3 applies also here. The only difference concerns the decoration: the topological defects of $\mathcal{Z}(\mathbb{A}^{(1)})$ related with the 1-form symmetry are surfaces, hence they have a different dimensionality of the lines generating the quantum symmetry $\mathsf{Rep}(G)$. Invariant lines, strictly speaking, do not come in copies labeled by representations but can be stacked with condensates of the quantum symmetry [97–99], and condensable algebras can sometimes be decorated with these condensates. We will not analyze this here, and we leave the analysis of the physical relevance of these decoration for the igSPT phases for future studies.

Starting from the condensable algebras for the 1-form symmetry $\mathbb{A}^{(1)}$ studied in section 3, we again have a classification into Type I, II and III igSPT phases. Interestingly, there are natural examples of all these types in (3+1)d. In particular, Type II igSPTs are gapless phases which would not be intrinsic in absence of the non-invertible symmetry, but the presence of the latter forbids to gap the theory, while Type III igSPTs are characterized by the presence of an anomalous, invertible duality symmetry in the infrared.

---

[32]This is a weaker condition because even if $-1$ is not a square, it might be a sum of two squares. For example, this is what happens for $n = 3$. However, for $n = 4$ even this weaker condition cannot be satisfied. In general, if $n = 0 \bmod(4)$ there is no solution, whereas for all other values of $n$ there is at least one.

**Type I igSPTs for $\mathbb{Z}_n \times \mathbb{Z}_n$ and $\phi_D$.** We look at theories with 1-form symmetry $\mathbb{Z}_n \times \mathbb{Z}_n$ and duality defects associated with the diagonal bicharacter $\phi_D(x, y) = (x, y)$. Clearly, interesting examples of igSPT phases are those that do not have a UV anomaly, so the theory would be compatible with a gapped SPT. Thus we look at integers $n \neq 0 \mod(4)$. As we have seen in section 3, the 1-form symmetry $\mathbb{Z}_n \times \mathbb{Z}_n$ admits igSPT phases only if $n$ can be written as the product of two non-coprime integers. The smallest such integer, which is $\neq 0 \mod(4)$ is $n = 9$, for which $p = q = 3$. The most general igSPT for the 1-form symmetry is determined by the condensable algebra

$$\mathcal{A}_{\mathbb{Z}_3 \times \mathbb{Z}_3, 1, \psi} = \left\{ \left( (3x, 3y); (\psi(3x, 3y)) \right) \mid x, y = 0, 1, 2 \right\}, \tag{124}$$

with $\psi$ represented by the matrix

$$\psi = \begin{pmatrix} s_1 & r_1 \\ s_2 & r_2 \end{pmatrix}, \tag{125}$$

with $r_1 \neq s_2 \mod(3)$ for the phase to be intrinsic. The condition of duality invariance is $\psi^2 = -1$, that is

$$s_1^2 + r_1 s_2 = r_2^2 + r_1 s_2 = -1, \qquad r_1(s_1 + r_2) = s_2(s_1 + r_2) = 0. \tag{126}$$

Setting $s_1 = r_2 = 0$ (since they represent stacking with an ordinary SPT), this can be solved by

$$s_2 = -1, \qquad r_1 = 1 \quad \text{(or vice versa)}. \tag{127}$$

Since $r_1 \neq s_2$, this leads to an algebra $\widehat{\mathcal{A}}_{\mathbb{Z}_3 \times \mathbb{Z}_3, 1, \psi}$ representing an igSPT phase for the full categorical structure including the duality defect.

**A Type II igSPT for $\mathbb{Z}_4 \times \mathbb{Z}_4$ and $\phi_O$.** Consider $\mathbb{A} = \mathbb{Z}_4 \times \mathbb{Z}_4$, and duality associated with the off-diagonal bicharacter $\phi_O(x, y) = (y, x)$. This symmetry is non-anomalous and compatible with gapped SPT phases.[33] It is easy to check that there are no Type I igSPTs in this case. There are, however, igSPT phases of Type II. Consider the subgroup $(\mathbb{Z}_2)_L \subset \mathbb{Z}_4 \times \mathbb{Z}_4$ with the homomorphism:

$$\psi(2a, 0) = (0, 2a). \tag{129}$$

The algebra $\mathcal{A}_0$ defined by this homomorphism

$$\mathcal{A}_0 = \{(0, 0; 0, 0), (2, 0; 0, 2)\}, \tag{130}$$

is duality invariant for the off-diagonal isomorphism. We have seen that for the 1-form symmetry this is a gSPT, but not igSPT. Indeed there are symmetric extensions, and the most general one is

$$\widehat{\psi}_{\sigma_1, \sigma_2, k}(x, y) = (2\sigma_1 x + (1 + 2\sigma_2)y, (1 + 2\sigma_2)x + ky), \tag{131}$$

where $\sigma_1, \sigma_2 = 0, 1$, while $k = 0, 1, 2, 3$. Since $\mathcal{A}_0$ is a duality invariant gSPT for the 1-form symmetry, it produces a gSPT $\widehat{\mathcal{A}}$ for the full categorical duality symmetry. To determine whether $\widehat{\mathcal{A}}$ is an igSPT (of Type II) or not we need to check if some of the extensions $\widehat{\psi}_{\sigma_1, \sigma_2, k}$ define duality invariant SPTs – if they do not, these are igSPTs. We have

$$\left( \phi_O^{-1} \widehat{\psi}_{\sigma_1, \sigma_2, k} \right)^2 = \begin{pmatrix} 1 + 2k\sigma_1 & 2k \\ 0 & 1 + 2k\sigma_1 \end{pmatrix}. \tag{132}$$

---

[33]There are duality invariant Lagrangian algebras for the 1-form symmetry, for instance:

$$\mathcal{L} = \{(x, -y; x, -y), \quad x, y = 0, \dots, 3\}. \tag{128}$$

For this to be $-1$, we would need $k$ to be even. But then $1 + 2k\sigma_1 = 1 \mod(4)$. We conclude that there is no duality-invariant symmetric extension of $\psi$. Hence $\widehat{\mathcal{A}}$ is an igSPT protected by the categorical duality symmetry.

**A complementary perspective: line order parameters.** Let us study the $\mathbb{Z}_4 \times \mathbb{Z}_4$ example from the perspective of line operators. The algebra $\mathcal{A}_0 = \{(0,0;0,0),(2,0;0,2)\}$ describes a gapless phase where the dyon

$$W_2^2 T_1^2 \,, \tag{133}$$

has perimeter law. The gapped dressing is duality-invariant with $\phi_O$ as the condensed line is. Now we ask whether we can condense the required dyons to reach a duality-invariant trivially gapped phase. Following our previous classification, for $n = 4$ these are described by $\alpha = 1,3 \,; \beta = 0$ and correspond to the condensation of

$$W_1 T_1 \,, \ W_2^3 T_2 \,, \quad \text{or} \quad W_1^3 T_1 \,, \ W_2 T_2 \,, \tag{134}$$

respectively. We notice, however, that all four of these lines are non-local w.r.t. $W_2^2 T_1^2$ and thus have area law. We conclude that the necessary monopole potential is inaccessible in the deep IR and the phase is protected by the duality symmetry.

Furthermore, we can also show that, breaking duality, it is possible to gap out the theory by an IR deformation. Consider the dyons $T_2 W_1$ and $T_1^{-1} W_2$, which are mutually local with respect to the condensed one and have perimeter law in the IR. We can turn on a monopole potential for them and let them condense. This corresponds to the completion

$$\mathcal{A} = \{(a,b;b,-a)\} \,, \tag{135}$$

for the algebra $\mathcal{A}_0$.

We can also be more specific about the realization $\mathcal{S}'$ of the symmetry in the gapless degrees of freedom and describe the anomaly. Condensing $\mathcal{A}_0$ gives rise to a $\mathbb{Z}_4 \times \mathbb{Z}_2$ DW theory, with lines generated by:

$$E = (0,1;1,0) \,, \qquad M = (0,0;0,1) \,, \tag{136}$$

$$E' = (1,0;0,1) \,, \qquad M' = (0,0;2,0) \,. \tag{137}$$

The UV duality symmetry acts by:

$$
\begin{aligned}
S(E) &= -E + M' \,, & S(M) &= M - E' \,, \\
S(E') &= -E' + 2M \,, & S(M') &= M' + 2E \,.
\end{aligned}
\tag{138}
$$

We can ask whether this duality symmetry is anomalous. The most general SPT for the 1-form symmetry is parametrized by algebras $\mathcal{L}_{l,s,s'}$ with generators:

$$E + lM + sM' \,, \quad E' + 2sM + s'M' \,. \tag{139}$$

We then ask whether any of these is duality-invariant. It is straightforward, although tedious, to show that none of these Lagrangian algebras are duality invariant. We thus conclude that the duality symmetry that acts on the gapless degrees of freedom is *anomalous*.

**A Type III igSPT for $\mathbb{Z}_2$.** Type III igSPTs are also quite easy to derive. Consider the simplest case $\mathbb{A} = \mathbb{Z}_2$, where an SPT for the duality symmetry is described by the dyonic algebra $\mathcal{L}_D$ generated by $(1,1)$. On the gapped boundary corresponding to $\mathcal{L}_D$ the symmetry is $\mathbb{Z}_2^{(0)} \times \mathbb{Z}_2^{(1)}$ with a mixed anomaly [1]:

$$I = \pi i \int A \cup \frac{1}{2}\mathfrak{P}(B) \,. \tag{140}$$

An equivariantization of $\mathcal{L}_D$ contains a choice of symmetry fractionalization class, $\eta \in H^2_\sigma(\mathbb{Z}_2, \mathbb{Z}_2) = \mathbb{Z}_2$. The non-trivial choice $\eta(A) = \beta(A) = \frac{1}{2}\delta A$ shifts the mixed anomaly and gives a term

$$I' = \pi i \int A \cup \beta(A)^2 \,, \tag{141}$$

describing an anomalous (invertible) duality symmetry. Thus, this algebra realizes a Type III igSPT from the non-invertible duality symmetry to $\mathbb{Z}_2^\omega$.

## Acknowledgments

We thank Fabio Apruzzi, Lakshya Bhardwaj, Thomas Dumitrescu, Dan Pajer, Alison Warman and Yunqin Zheng for discussions.

**Funding information**  A.A. is supported by the ERC-COG grant NP-QFT No. 864583 "Non-perturbative dynamics of quantum fields: from new deconfined phases of matter to quantum black holes", by the MUR-FARE2020 grant No. R20E8NR3HX "The Emergence of Quantum Gravity from Strong Coupling Dynamics", by the MUR-PRIN2022 grant No. 2022NY2MXY, and by the INFN "Iniziativa Specifica ST&FI". C.C. is supported by STFC grant ST/X000761/1. The work of SSN is supported by the UKRI Frontier Research Grant, underwriting the ERC Advanced Grant "Generalized Symmetries in Quantum Field Theory and Quantum Gravity".

## A  Drifeld center of $\mathsf{Vec}_{D_8}$

$D_8$ can be realized as a semidirect product $\mathbb{Z}_4 \rtimes \mathbb{Z}_2$, where $G = \mathbb{Z}_2$ acts as $a \mapsto -a$ on $\mathbb{Z}_4$. Hence $\mathfrak{Z}(\mathsf{Vec}_{D_8})$ can be obtained following the procedure described in section 4.1.1, starting from $\mathfrak{Z}(\mathsf{Vec}_{\mathbb{Z}_4})$ and gauging the $G = \mathbb{Z}_2$ symmetry acting as charge conjugation

$$C : (a, b) \mapsto (-a, -b), \quad (a, b) \in \mathfrak{Z}(\mathsf{Vec}_{\mathbb{Z}_4}) = \mathbb{Z}_4 \times \mathbb{Z}_4 \,. \tag{A.1}$$

The invariant lines, that coincide with their own orthogonal (with respect to the braiding), are

$$F = F^\perp = \{(2x, 2y) \mid x, y = 0, 1\} \cong \mathbb{Z}_2 \times \mathbb{Z}_2 \,. \tag{A.2}$$

The twist defects in the $G$-crossed extension of $\mathfrak{Z}(\mathsf{Vec}_{\mathbb{Z}_4})$ can be labelled as

$$\sigma_{(a,b)}, \qquad (a, b) \in \{(0, 0), (1, 0), (0, 1), (1, 1)\} = \frac{\mathbb{Z}_4 \times \mathbb{Z}_4}{F^\perp} \,. \tag{A.3}$$

Notice that all twist defects are charge conjugation invariant.

Now we gauge $G = \mathbb{Z}_2$. We will denote by $\pm$ the trivial and non-trivial representations of $\mathbb{Z}_2$. From the invariant bulk lines we obtain abelian anyons:

$$L^\pm_{0,0}, \quad L^\pm_{2,0}, \quad L^\pm_{0,2}, \quad L^\pm_{2,2} \,. \tag{A.4}$$

From the non-invariant bulk anyons we get the dimension-two anyons:

$$L_{1,0}, \quad L_{0,1}, \quad L_{1,1}, \quad L_{2,1}, \quad L_{1,2}, \quad L_{1,3} \,. \tag{A.5}$$

Finally since the twist defects are all charge conjugation invariant we get

$$\Sigma^\pm_{0,0}, \quad \Sigma^\pm_{1,0}, \quad \Sigma^\pm_{0,1}, \quad \Sigma^\pm_{1,1} \,. \tag{A.6}$$

In table 11 we present the translation between these labels and other labels for $\mathcal{Z}(\mathsf{Vec}_{D_8})$ used in the literature [38].

Table 11: The table lists all anyons of $\mathfrak{Z}(\mathsf{Vec}_{D_8})$ using three distinct notations, see [38]. The first column uses our notation, which is natural in the context of gauging. The second column employs the standard notation for $\mathfrak{Z}(\mathsf{Vec}_G)$ for any finite group $G$, expressed through conjugacy classes $[g]$ and stabilizer representations $\rho$. The third column shows the corresponding labels in terms of three copies of the toric code, as referenced in [100]. The final two columns display the quantum dimension and spin.

| Outer Auto | $([g], \rho)$ | Anyon label | Dim | $T$ |
|---|---|---|---|---|
| $L_{0,0}^+$ | $(1,1)$ | $1$ | $1$ | $1$ |
| $L_{0,0}^-$ | $(1,1_a)$ | $e_{RG}$ | $1$ | $1$ |
| $L_{0,2}^-$ | $(1,1_x)$ | $e_R$ | $1$ | $1$ |
| $L_{0,2}^+$ | $(1,1_{ax})$ | $e_G$ | $1$ | $1$ |
| $L_{0,1}$ | $(1,E)$ | $m_B$ | $2$ | $1$ |
| $L_{2,0}^-$ | $(a^2,1)$ | $e_{RGB}$ | $1$ | $1$ |
| $L_{2,0}^+$ | $(a^2,1_a)$ | $e_B$ | $1$ | $1$ |
| $L_{2,2}^+$ | $(a^2,1_x)$ | $e_{GB}$ | $1$ | $1$ |
| $L_{2,2}^-$ | $(a^2,1_{ax})$ | $e_{RB}$ | $1$ | $1$ |
| $L_{2,1}$ | $(a^2,E)$ | $f_B$ | $2$ | $-1$ |
| $L_{1,0}$ | $(a,1)$ | $m_{RG}$ | $2$ | $1$ |
| $L_{1,1}$ | $(a,i)$ | $s_{RGB}$ | $2$ | $i$ |
| $L_{1,2}$ | $(a,-1)$ | $f_{RG}$ | $2$ | $-1$ |
| $L_{1,3}$ | $(a,-i)$ | $\bar{s}_{RGB}$ | $2$ | $-i$ |
| $\Sigma_{1,1}^+$ | $(x,+,+)$ | $m_{GB}$ | $2$ | $1$ |
| $\Sigma_{1,0}^+$ | $(x,+,-)$ | $m_G$ | $2$ | $1$ |
| $\Sigma_{1,0}^-$ | $(x,-,-)$ | $f_G$ | $2$ | $-1$ |
| $\Sigma_{1,1}^-$ | $(x,-,+)$ | $f_{GB}$ | $2$ | $-1$ |
| $\Sigma_{0,1}^+$ | $(ax,+,+)$ | $m_{RB}$ | $2$ | $1$ |
| $\Sigma_{0,0}^+$ | $(ax,+,-)$ | $m_R$ | $2$ | $1$ |
| $\Sigma_{0,0}^-$ | $(ax,-,-)$ | $f_R$ | $2$ | $-1$ |
| $\Sigma_{0,1}^-$ | $(ax,-,+)$ | $f_{RB}$ | $2$ | $-1$ |

# B  igSPTs for 2-groups in (2+1)d

In this appendix we briefly discuss igSPTs for 2-group symmetries in $(2+1)$-dimensions. Consider a discrete 2-group $\Gamma$, with 0-form symmetry $\mathbb{A}$ and a 1-form symmetry $\mathbb{B}$:

$$1 \longrightarrow \mathbb{B} \longrightarrow \Gamma \longrightarrow \mathbb{A} \longrightarrow 1. \tag{B.1}$$

We consider a trivial $\mathbb{A}$ action on $\mathbb{B}$ *but* a non-trivial Postinikov class $\beta \in H^3(\mathbb{A}, \mathbb{B})$ [101]. The SymTFT for this system admits a Lagrangian description[34]

$$S = 2\pi i \int_{4d} C \cup \delta B + 2\pi i \int_{4d} T \cup \delta A + 2\pi i \int_{4d} C \cup \beta(A), \tag{B.2}$$

with $C \in C^1(X, \mathbb{B}^\vee)$ and $T \in C^2(X, \mathbb{A}^\vee)$, while $B \in C^2(X, \mathbb{B})$ and $A \in C^1(X, \mathbb{A})$. Integrating out $C$ imposes the 2-group constraint:

$$\delta B = -\beta(A). \tag{B.3}$$

The 2-group is realized by imposing Dirichlet boundary conditions for $B$ and $A$, which become the background fields $B$ and $A$ for the 2-group. Now suppose that, as groups, $\mathbb{B}$ is a subgroup of $\mathbb{A}^\vee$, and let $\iota : \mathbb{B} \hookrightarrow \mathbb{A}^\vee$ be the corresponding embedding. This induces a (surjective) map $\iota^\vee : \mathbb{A} \to \mathbb{B}^\vee$ defined by $\iota^\vee(a)(b) = \iota(b) \cdot a$.

This allows us to define a new interface $\mathcal{I}_\iota$, by: $C - \iota^\vee(A) = 0$, or, in terms of lines, by condensing the algebra generated by

$$\exp\left(2\pi i \int_\gamma (C - \iota^\vee(A))\right). \tag{B.4}$$

Since these lines braid non-trivially with the surfaces $e^{i\int_\Sigma B}$, the reduced topological order is

$$S_{\mathcal{I}_\iota} = 2\pi i \int_{4d} T \cup \delta A + 2\pi i \int_{4d} \iota^\vee(A) \cup \beta(A). \tag{B.5}$$

This is a Dijkgraaf-Witten theory for $\mathbb{A}$, with twist $\omega \in H^4(\mathbb{A}, U(1))$ such that

$$A^*(\omega) = \iota^\vee(A) \cup \beta(A). \tag{B.6}$$

Physically we are describing a situation where the lines charged under the 1-form symmetry part of the 2-group are completely confined, and there is a gapless theory capturing the low energy dynamics below the scale of confinement. The 1-form symmetry is trivialized in the IR, while the 0-form symmetry $\mathbb{A}$ acts on the gapless sector. If $\omega \in H^4(\mathbb{A}, U(1))$ is non-trivial, the 0-form symmetry has an emergent anomaly and the gapless phase is an igSPT.

A simplest example is $\mathbb{B} = \mathbb{Z}_n$ and $\mathbb{A} = \mathbb{Z}_n \times \mathbb{Z}_n$, with $\iota : \mathbb{Z}_n \to \mathbb{Z}_n \times \mathbb{Z}_n$ the embedding in the right factor.[35] Decomposing $A = (A_1, A_2)$ as a pair of two $\mathbb{Z}_n$ gauge fields, we have $\iota^\vee(A) = A_2$. If the Postnikov class is such that

$$\beta(A) = A_1 \cup \mathrm{Bock}(A_1), \tag{B.8}$$

then the emergent anomaly for the IR symmetry $\mathbb{Z}_n \times \mathbb{Z}_n$ is non-trivial

$$I = 2\pi i \int A_2 \cup A_1 \cup \mathrm{Bock}(A_1). \tag{B.9}$$

---

[34]Here $\beta(a) \in C^3(X, \mathbb{B})$ denotes the pull-back $a^*(\beta)$, with $a$ realized as a map $X \to B\mathbb{A}$.

[35]Anomalies for $\mathbb{Z}_n \times \mathbb{Z}_n$ in 3d are classified by $H^4(\mathbb{Z}_n \times \mathbb{Z}_n, U(1)) \cong \mathbb{Z}_n \times \mathbb{Z}_n$. The two generators are

$$2\pi i \int_{4d} A_1 \cup A_2 \cup \mathrm{Bock}(A_2), \qquad 2\pi i \int_{4d} A_2 \cup A_1 \cup \mathrm{Bock}(A_1). \tag{B.7}$$

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
