# Peer review of "SymTFT for (3+1)d Gapless SPTs and Obstructions to Confinement"

_SciPost Physics, doi:SciPost Phys. 18, 114 (2025)_

## Round 2 · Referee Report · Anonymous (Referee 1) · 2025-1-27

Report
This paper generalizes the notion of igSPT in 2-dim to igSPT in 4-dim and discuss the three types of igSPT of G-extension of Abelian symmetries in both 2-dim and 4-dim. It is a solid progress in the study of categorical symmetries and reveals some interesting future research directions. The paper is well-written with a nice combination of generic result and concrete examples.
However, before I can recommend the manuscript for publication, there are a few questions and comments to the authors as well as some potential typos to be corrected:
-
In Section 1.1 when discussing the possible Gapped symmetry boundary, the author claims that the symmetry boundary is "specified in terms of which topological defects can end on this boundary". In the recent work (https://arxiv.org/abs/2501.03314) it is pointed out that there exists two Lagrangian algebras distinguished only by the multiplication maps, which implies that the corresponding gapped boundaries have the same set of endable defects, and yet are different. The author might want to clarify this to give a more precise statement.
-
In section 4.1.1 when discussing the structure of $Aut_0(\mathbb{A} \times \mathbb{A}^\vee)$, the author claims that this group is isomorphic to $Aut(\mathbb{A}) \rtimes \mathbb{Z}_2$, but I think this is not true unless $\mathbb{A} = \mathbb{Z}_N$. The generators of this group for generic $\mathbb{A}$ is discussed in Section 4 of https://arxiv.org/pdf/1404.6646, and the subgroup which preserves the anyons in $\mathbb{A}$ but maps anyons in $\mathbb{A}^\vee$ to dyons are missing here. From the boundary point of view, they corresponds to dress A-SPTs.
-
In eq (4.19), it is not obvious where $\chi$ is defined (perhaps the authors mean $\gamma_O$) as well as the meaning of $\chi[\psi]$. It would be nice if the authors could clarify this in the manuscript.
And there are a few non-substantial typos and suggestions:
-
In footnote 4, the authors use $d$ for differential while above the eq (1.2) $\delta$ is used instead. Throughout the paper, both notations are used and perhaps it is good to be consistent, or at least clarify if there is any difference between two notations.
-
In Section 2.3.1, in the paragraph there are two places where the image of $\psi$ and $\widehat{\psi}$ should strictly speaking be $\mathbb{Z}_4^\vee$ instead of $\mathbb{Z}_4$.
-
In eq (2.26), should the $\ell$ be $x$ instead? Furthermore, since this is igSPT, k = 0 should also be excluded?
-
In eq (3.28), the $a$ on the RHS should be $x$ instead.
-
In footnote 27, in "Type I igSPTs realize anomalous 1-form symmetries in the IR", what is this 1-form symmetry (in the 2d theory) or it is actually 0-form symmetries?
Recommendation
Ask for minor revision

---

## Round 2 · Referee Report · Anonymous (Referee 2) · 2025-2-1

Report
Apart from the comments from the first report, the referee finds it interesting that the minimal igSPT in 3+1d is for symmetry $\mathbb{A}= \mathbb{Z}_4\times \mathbb{Z}_2$, rather than $\mathbb{A}= \mathbb{Z}_4$ (as in 1+1d). This is related to the fact that in 1+1d there is a mod 2 anomaly for $\mathbb{Z}_2$ symmetry (serving as the "low energy sector anomaly"), while in 3+1d the mod 2 anomaly for $\mathbb{Z}_2$ 1-form symmetry can not be detected on spin manifold (as assumed throughout the paper). But once we put the theory on non-spin manifold, there is a igSPT, protected by gravity as well as $\mathbb{Z}_2$ symmetry. The referee finds it helpful to emphasize this point, to make further contrast between 1+1d and 3+1d.
Recommendation
Publish (easily meets expectations and criteria for this Journal; among top 50%)

---

## Round 3 · Referee Report · Anonymous (Referee 1) · 2025-3-3

Report

This paper generalizes the notion of igSPT in 2-dim to igSPT in 4-dim and discuss the three types of igSPT of G-extension of Abelian symmetries in both 2-dim and 4-dim. It is a solid progress in the study of categorical symmetries and reveals some interesting future research directions. The paper is well-written with a nice combination of generic result and concrete examples. I recommend the paper to be published.

Recommendation

Publish (easily meets expectations and criteria for this Journal; among top 50%)

---

## Round 3 · Author Response

We thank both referees for the useful comments on the paper.

In particular the point raised by referee 2 is very interesting as it open the possibility of realizing new igSPTs in dimension (3+1)d, that can only be detected on non-spin manifolds. We commented about this at the end of Section 3.3.2. However we couldn't find a clear physical application, and we leave this interesting problem for the future.

We also agree with all the comments of referee 1.

---

## Round 3 · List of Changes

More in detail:

  1. We added footnote 8 to clarify this point. The referee is indeed absolutely right, but in the specific examples discussed in our paper this issue never arises, so we use the expression "Lagrangian algebra" to mean "Lagrangian algebra object".

  2. The referee is absolutely right, and we made substantial changes in the paragraph "Structure of $ Aut(A x \mathbb{A}^\vee)$". As this was never really used, the previous mistake does not propagate in the rest of the paper.

  3. Corrected.

We also corrected all the typos and suggestions.

---

## Editorial Decision

published